# TRANSFORMERS LEARN LATENT MIXTURE MODELS IN-CONTEXT VIA MIRROR DESCENT

**Francesco D'Angelo**[*]
TML Lab, EPFL
Switzerland

**Nicolas Flammarion**
TML Lab, EPFL
Switzerland

## ABSTRACT

Sequence modelling requires determining which past tokens are causally relevant from the context and their importance: a process inherent to the attention layers in transformers, yet whose underlying learned mechanisms remain poorly understood. In this work, we formalize the task of estimating token importance as an in-context learning problem by introducing a framework based on Mixture of Transition Distributions, where a latent variable determines the influence of past tokens on the next. The distribution over this latent variable is parameterized by unobserved mixture weights that transformers must learn in-context. We demonstrate that transformers can implement Mirror Descent to learn these weights from the context. Specifically, we give an explicit construction of a three-layer transformer that exactly implements one step of Mirror Descent and prove that the resulting estimator is a first-order approximation of the Bayes-optimal predictor. Corroborating our construction and its learnability via gradient descent, we empirically show that transformers trained from scratch learn solutions consistent with our theory: their predictive distributions, attention patterns, and learned transition matrix closely match the construction, while deeper models achieve performance comparable to multi-step Mirror Descent.

## 1 INTRODUCTION

In recent years, Machine Learning has been transformed by large language models (LLMs). These massive, complex models achieve unprecedented performance across diverse tasks beyond text generation (Bubeck et al., 2023). A striking example is in-context learning (ICL) (Brown et al., 2020; Min et al., 2022), where models adapt to new tasks using only examples in the prompt without parameter updates. Mechanistic interpretability has made significant strides in explaining this phenomenon, revealing that transformers can implement computational circuits (Elhage et al., 2021; Olsson et al., 2022; D'Angelo et al., 2025) that mimic known algorithms. For instance, in settings like linear regression, they learn to implement gradient-based optimization (Garg et al., 2022; Akyürek et al., 2022; Von Oswald et al., 2023a;b; Zhang et al., 2023; Ahn et al., 2023; Mahankali et al., 2024), while for Markov Chains, they implement counting-based estimators for the transition probabilities (Nichani et al., 2024; Edelman et al., 2024; Bietti et al., 2023a; Rajaraman et al., 2024; Ildiz et al., 2024; Svete & Cotterell, 2024; Chen et al., 2024). However, these successes are confined to problems where all the sequences rely on the same, fixed causal structure: the relationship between tokens is static. For instance, in the case of regression, the model only needs to learn that every even token depends on the previous odd token and in Markov chains, it learns that the next token depends only on the previous one.

Real-world sequential data, particularly language, defies such simplicity. The meaning of a sentence arises not from the fixed sequence of word meanings, but from the dynamic causal links between the words that must be inferred from the context. These underlying structures, which are fundamental to language, are latent variables hidden from direct observation. Consider the sentence in Figure 1; to predict the final token, a model cannot rely on simple recency. It must infer a latent structure: that "dog" is the agent, "ball" is the relevant object, and "bird" is a distractor. The influence of a past token is not merely a function of its position, but of its inferred role. This ability to infer and reason

---

[*]Correspondence to: `francesco.dangelo@epfl.ch`

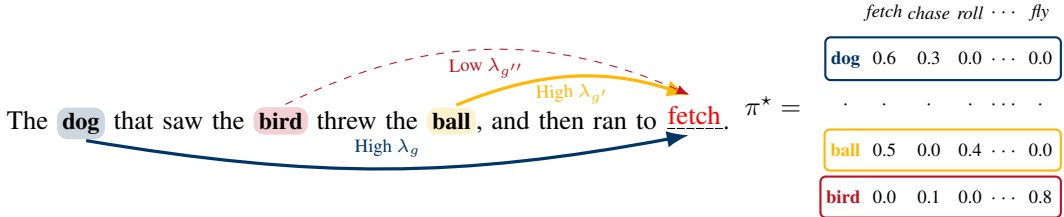

Figure 1: To predict the final word, the model must infer the causal relevance of past tokens. Our MTD framework models this by separating a static, context-free unigram ($\pi^\star$) from dynamic, context-dependent weights ($\boldsymbol{\lambda}$) that are inferred in-context. The model learns to assign high weights to the causally relevant positions ('dog') and ('ball'), activating their respective slices of the unigram (e.g., $\pi^\star(\text{dog}, \cdot)$ favouring verbs like *fetch* or *chase*). Conversely, a low weight is assigned to the distractor ('bird'), suppressing its influence.

over unobserved variables, be it syntactic roles, speaker intent, or the causal topic of a discourse, is a hallmark of intelligence. A robust model must therefore move beyond fixed dependencies and learn to infer this latent structure, dynamically identifying which tokens are causally relevant in a given context. These considerations give rise to the core question of this paper:

*Can transformers infer latent structures in-context, and what algorithm do they learn?*

To investigate this question, we introduce a synthetic task based on the Mixture of Transition Distributions (MTD) model (Raftery, 1985; André Berchtold, 2002). This framework recasts token-importance estimation as learning latent mixture weights in-context. While prior work has used mixture models in regressions (Pathak et al., 2024) or HMMs (Xie et al., 2022) to study ICL, they did not unveil the mechanism through which transformers learn to infer the latent mixture weights. We make the following contributions:

1. We introduce a framework for ICL based on latent variables using MTD: we create a synthetic task that frames the estimation of the influence of past tokens as learning latent mixture weights in-context.

2. We show that transformers can solve this task by implementing Mirror Descent and give an explicit construction of a 3-layer transformer that exactly implements one step of this algorithm.

3. We prove that the resulting one-step estimator yields a first-order approximation of the Bayes-optimal predictor.

4. We empirically validate that transformers trained from scratch with Adam learn solutions consistent with this construction matching the predictive distributions and attention patterns while deeper models achieve performance comparable to multi-step Mirror Descent.

We show that transformers implement Mirror Descent to dynamically infer which past tokens are relevant, a capability fundamental to sequence modeling and, in particular, to capturing higher-order dependencies in language. This provides a gradient-based explanation of ICL in sequence modeling and offers a new algorithmic lens for understanding latent-variable inference in attention-based models. We defer to the Appendix C a more detailed discussion of related work.

**Why MTD?** Our choice of the MTD model is motivated by the desire to capture both in-weight and in-context learning in a single, controlled setting. While prior ICL work has focused on settings and tasks where nearly all useful structure must be inferred from the prompt (e.g., gradient-descent ICL in linear regression or counting estimators for simple Markov chains), our setup explicitly assumes that some statistical structure, namely the transition matrix $\pi^\star$, can be stored in the model's weights during pretraining and reused at inference time to infer in-context the mixture weights $\boldsymbol{\lambda}$ from a single sequence. This mirrors a plausible regime for large language models, which cannot memorize full high-order $n$-grams due to their sample complexity growing exponentially with $n$ but can realistically encode lower-order $n$-grams in their weights and dynamically reweight the influence of different past tokens depending on the context. Our in-context learning framework explicitly couples an in-weight component (learning $\pi^\star$) with an in-context component (inferring $\boldsymbol{\lambda}$), providing a natural testbed for studying the interplay between in-weight and in-context learning in LLMs.

## 2 DISENTANGLED TRANSFORMERS

The *disentangled transformer* (Friedman et al., 2023) is a modification of a standard Transformer with relative positional encodings (RPE) (Shaw et al., 2018), designed for enhanced interpretability by: (1) removing MLPs; (2) replacing residual connections with concatenation, creating an explicit residual stream that preserves computations from all previous layers; and (3) simplifying the attention mechanism to use a single attention matrix instead of separate query and key matrices, and absorbing the value projection into the output matrix. Nichani et al. (2024) demonstrated that disentangled transformers are equivalent to standard transformers using only attention layers. The model maps a sequence of tokens $\boldsymbol{s} = (s_1, \dots, s_T)$ from a finite alphabet $\mathcal{S}$ to a sequence of vectors. Each token $s_i$ is represented by its one-hot vector $\boldsymbol{e}_{s_i} \in \{0,1\}^{|\mathcal{S}|}$, yielding the initial representation $\boldsymbol{h}_i^{(0)} = \boldsymbol{e}_{s_i} \in \mathbb{R}^{d_0}$ with $d_0 = |\mathcal{S}|$. The model consists of $L$ layers. We denote by $\boldsymbol{h}_i^{(l)} \in \mathbb{R}^{d_l}$ the full hidden state of token $i$ after layer $l$, and by $\hat{\boldsymbol{h}}_i^{(l,h)}$ the output of attention head $h$ in layer $l$. For each head $h \in \{1, \dots, H_l\}$, the pre-softmax attention score $e_{ij}^{(l,h)}$ between query $i$ and key $j$ is:

$$e_{ij}^{(l,h)} = (\boldsymbol{h}_i^{(l-1)})^\top \boldsymbol{W}_A^{(l,h)} \boldsymbol{h}_j^{(l-1)} + (\boldsymbol{h}_i^{(l-1)})^\top \boldsymbol{r}_{ij}^{(l,h)}. \tag{1}$$

Here, $\boldsymbol{W}_A^{(l,h)} \in \mathbb{R}^{d_{l-1} \times d_{l-1}}$ is a learnable attention matrix and $\boldsymbol{r}_{ij}^{(l,h)} = (\boldsymbol{R}_A^{(l,h)})_{i-j+1,:}$ is a relative positional encoding vector retrieved from a learnable lookup table $\boldsymbol{R}_A^{(l,h)} \in \mathbb{R}^{T \times d_{l-1}}$. The attention weights are computed via a causally masked softmax: $\mathcal{A}_{ij}^{(l,h)} = [\text{softmax}(\boldsymbol{e}_i^{(l,h)})]_j$, where $\boldsymbol{e}_i^{(l,h)}$ is the vector of scores. The output of a single attention head is formed by concatenating the full hidden state $\boldsymbol{h}_j^{(l-1)}$ with a positional value embedding $\boldsymbol{r'}_{ij}^{(l,h)}$ before the weighted sum. The full hidden state is then updated by concatenating the input with all head outputs:

$$\hat{\boldsymbol{h}}_i^{(l,h)} = \sum_{j=1}^T \mathcal{A}_{ij}^{(l,h)} \text{Concat}\left(\boldsymbol{h}_j^{(l-1)}, \boldsymbol{r'}_{ij}^{(l,h)}\right) \quad \boldsymbol{h}_i^{(l)} = \text{Concat}\left(\boldsymbol{h}_i^{(l-1)}, \hat{\boldsymbol{h}}_i^{(l,1)}, \dots, \hat{\boldsymbol{h}}_i^{(l,H_l)}\right).$$

The positional value embeddings $\boldsymbol{r'}_{ij}^{(l,h)}$ are retrieved from a second lookup table $\boldsymbol{R}_V^{(l,h)} \in \mathbb{R}^{T \times d_R}$, where $d_R$ is a fixed hyperparameter. The dimension of the representation thus grows according to the recurrence $d_l = d_{l-1} + H_l \cdot (d_{l-1} + d_R)$. After $L$ layers, a final linear layer with matrix $\boldsymbol{W}_O \in \mathbb{R}^{|\mathcal{S}| \times d_L}$ maps $\boldsymbol{h}_i^{(L)}$ to logit predictions.

## 3 MIXTURE OF TRANSITION DISTRIBUTIONS FOR IN-CONTEXT LEARNING

The Mixture of Transition Distributions (MTD) model (Raftery, 1985), is a higher-order Markov chain that offers a parsimonious representation of long-range dependencies. The idea is to model the probability of the next state as a mixture over lags, where a transition matrix is applied to the token at each lag, and the mixture weights determine the relative influence of each past position.

**Model Definition:** Let $\boldsymbol{Y} = (Y_1, \dots, Y_T)$ be a sequence of random variables taking values in a finite alphabet $\mathcal{Y} = \{1, \dots, q\}$. The MTD model of order $m$ explains this sequence by positing a corresponding sequence of unobserved latent variables $\boldsymbol{Z} = (Z_{m+1}, \dots, Z_T)$, where each $Z_t \in \{1, \dots, m\}$ acts as a switch, selecting which of the $m$ previous tokens, $Y_{t-1}, \dots, Y_{t-m}$, will influence the current token $Y_t$. The selection of this lag is a random event, governed by the mixture weights $\boldsymbol{\lambda} = (\lambda_1, \dots, \lambda_m)$ with $\lambda_g \geq 0$ for all $g$ and $\sum_{g=1}^m \lambda_g = 1$, such that the probability of choosing lag $g$ is given by $\mathbb{P}(Z_t = g) = \lambda_g$. Once the lag $Z_t = g$ is sampled, the next token $Y_t$ is generated from a first-order transition that depends only on the state at the sampled position, $Y_{t-g}$. This is captured by a transition matrix $\boldsymbol{\pi} \in \mathcal{P}$ with $\mathcal{P}$ the set of $q \times q$ row-stochastic matrices defining the conditional probabilities $\pi(i,j) = \mathbb{P}(Y_t = j \mid Y_{t-g} = i, Z_t = g)$. Marginalizing over the unobserved latent variable $Z_t$, we obtain the predictive distribution:

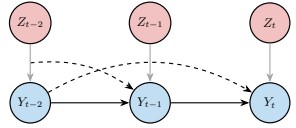

Figure 2: MTD for m=2.

**Definition 1** (Mixture Transition Distribution). *A sequence of random variables $\boldsymbol{Y}$ follows an $m$-th order MTD model if for all $t > m$ and any history $\boldsymbol{y}_1^{t-1}$, the conditional probability of $Y_t$ is:*

$$\mathbb{P}(Y_t = y_t \mid \boldsymbol{Y}_1^{t-1} = \boldsymbol{y}_1^{t-1}, \boldsymbol{\lambda}) = \sum_{g=1}^m \lambda_g \, \pi(y_{t-g}, y_t), \tag{2}$$

*where the parameters $\boldsymbol{\lambda} = (\lambda_1, \dots, \lambda_m)$ are the mixture weights and $\boldsymbol{\pi}$ the transition matrix.*

This structure allows one to model different effective contexts, making it more flexible than a first-order Markov chain while preserving tractability by requiring only $m - 1 + q(q - 1)$ parameters compared to the $q^m(q - 1)$ of a full $m$-th order Markov chain.

**The In-Context Learning Task:** We design an in-context learning task based on the MTD model structured as follows: Given a transition matrix $\boldsymbol{\pi}$ we generate sequences $\boldsymbol{y}$, by sampling for each a new vector of mixture weights $\boldsymbol{\lambda}$ from a prior $\boldsymbol{\lambda} \sim \text{Dirichlet}(\boldsymbol{\alpha} = \mathbf{1})$. This $\boldsymbol{\lambda}$ vector is the hidden *task* that defines the statistical structure of that particular sequence. The sequence $\boldsymbol{y} = (y_1, \ldots, y_T)$ is then generated according to the MTD model in equation 2. The learning objective for a model $f \in \mathcal{F}$, such as a transformer, is to predict the next token $y_t$ given the context $\boldsymbol{y}_1^{t-1}$:

$$\inf_{f \in \mathcal{F}} \mathbb{E}_{\boldsymbol{\lambda} \sim \text{Dirichlet}(\boldsymbol{\alpha})} \mathbb{E}_{\boldsymbol{y} \sim \text{MTD}(\boldsymbol{\lambda}, \boldsymbol{\pi}^\star)} \left[ \text{KL}\left( p(Y_t | \boldsymbol{y}_1^{t-1}, \boldsymbol{\lambda}) \, \| \, f(\boldsymbol{y}_1^{t-1}) \right) \right]. \tag{3}$$

To perform this prediction optimally, the model must effectively learn the mixture weights $\boldsymbol{\lambda}$ of the latent variables $Z_t$, which are not directly observable in the sequence and differ across sequences:

> *In-Context Task*: Learn the unknown mixture weights $\boldsymbol{\lambda}$ given a single sequence $\boldsymbol{y}$.

**The Bayes Optimal Solution:** The solution to the ICL task (Eq. 3) is the Bayesian predictive distribution, $p(Y_{t+1} \mid \boldsymbol{y}_1^t, \boldsymbol{\alpha})$.

**Proposition 1** (Bayes-Optimal MTD Predictor). *Given the MTD model with a known transition matrix $\boldsymbol{\pi}$, a prior $p(\boldsymbol{\lambda}|\boldsymbol{\alpha})$, and an observed sequence $\boldsymbol{y}_1^t$, the Bayesian predictive distribution for $Y_{t+1} = j$ is a convex combination of the lag-specific transition probabilities, weighted by the posterior mean of the mixture weights $\boldsymbol{\lambda}$:*

$$p(Y_{t+1} = j \mid \boldsymbol{y}_1^t, \boldsymbol{\alpha}) = \sum_{g=1}^m \hat{\lambda}_g^{Bayes} \cdot \pi(y_{t+1-g}, j) \quad \hat{\lambda}_g^{Bayes} := \mathbb{E}[\lambda_g \mid \boldsymbol{y}_1^t, \boldsymbol{\alpha}] = \int_{\Delta_{m-1}} \lambda_g \cdot p(\boldsymbol{\lambda} \mid \boldsymbol{y}_1^t, \boldsymbol{\alpha}) \, d\boldsymbol{\lambda},$$

$$\tag{4}$$

*where $p(\boldsymbol{\lambda} \mid \boldsymbol{y}_1^t, \boldsymbol{\alpha}) \propto p(\boldsymbol{y}_1^t \mid \boldsymbol{\lambda}) p(\boldsymbol{\lambda} \mid \boldsymbol{\alpha})$ is the posterior distribution with $p(\boldsymbol{y}_1^t \mid \boldsymbol{\lambda}) = \prod_{k=m+1}^t \left( \sum_{h=1}^m \lambda_h \pi(y_{k-h}, y_k) \right)$ being the likelihood.*

The derivation is provided in Appendix E. While elegant, the Bayes-optimal predictor is analytically intractable. The core issue is that the Dirichlet prior is not conjugate to the MTD likelihood, meaning the posterior distribution does not have a closed form. Consequently, the integral defining the posterior mean $\hat{\lambda}_g^{\text{Bayes}}$ cannot be computed directly, necessitating the use of approximation methods.

**Mirror Descent (MD):** The intractability of the posterior mean motivates considering simpler point estimates, such as the Maximum Likelihood Estimate (MLE) or the Maximum A Posteriori (MAP) estimate which are equivalent under uniform Dirichlet prior. However, analytical computation of either the MLE or MAP is intractable, necessitating iterative optimization methods. Methods such as Expectation–Maximization (EM) or Mirror Descent (MD) are preferred over standard gradient descent, as they are naturally adapted to the geometry of the simplex. For an extended discussion, see Appendix G. Furthermore, even if the MAP estimate could be found, it represents the mode of the posterior distribution, which does not coincide with the posterior mean in the Bayes-optimal predictor, therefore leading to suboptimal predictions.

Mirror Descent, is a first-order method particularly well-suited for optimizing over the probability simplex $\Delta_{m-1}$ Nemirovskij & Yudin (1983); Beck & Teboulle (2003). By using a Bregman divergence based on the negative entropy potential $\Psi(\boldsymbol{\lambda}) = -H(\boldsymbol{\lambda})$, where $H(\boldsymbol{\lambda}) = -\sum_{g=1}^m \lambda_g \log \lambda_g$ is the Shannon entropy, MD results in the **Exponentiated Gradient (EG)** algorithm, which has a simple multiplicative update rule (see Appendix G.4 for details):

$$\lambda_g^{(k+1)} = \frac{\lambda_g^{(k)} \exp(\eta \cdot \nabla_\lambda \ell(\boldsymbol{\lambda}^{(k)})_g)}{\sum_{h=1}^m \lambda_h^{(k)} \exp(\eta \cdot \nabla_\lambda \ell(\boldsymbol{\lambda}^{(k)})_h)}, \tag{5}$$

where $\eta > 0$ is the learning rate and $\nabla_\lambda \ell(\boldsymbol{\lambda}^{(k)})$ the gradient of the log-likelihood evaluated at $\boldsymbol{\lambda}^{(k)}$. Instead of iterating the EG algorithm to convergence, we analyze a non-iterative estimator derived from a single update step, initialized at the center of the probability simplex ($\boldsymbol{\lambda}^{(0)} = (1/m, \ldots, 1/m)$). This approach yields a computationally efficient, regularized approximation of the MLE, which we demonstrate serves as an effective proxy for the posterior mean.

**Proposition 2** (One-Step MD Estimator)**.** *Initializing the EG algorithm (Eq. 5) at $\boldsymbol{\lambda}^{(0)} = (1/m, \ldots, 1/m)$ and applying a single update step for the MTD model yields the estimator:*

$$
\hat{\lambda}_g^{MD} := \frac{\exp\left(\eta \cdot m \sum_{k=m+1}^{t} \gamma_k(g)\right)}{\sum_{j=1}^{m} \exp\left(\eta \cdot m \sum_{k=m+1}^{t} \gamma_k(j)\right)} \quad \gamma_k(g) := p(Z_k = g \mid \boldsymbol{y}_1^k, \boldsymbol{\lambda}^{(0)}) = \frac{\pi(y_{k-g}, y_k)}{\sum_{h=1}^{m} \pi(y_{k-h}, y_k)},
$$

(6)

*where $\gamma_k(g)$ is the posterior responsibility of lag $g$ at step $k > m$, under the uniform prior.*

The derivation is provided in Appendix G.4. This one-step estimator computes, for each step $k$ in the sequence, the posterior probability that lag $g$ was responsible for generating $y_k$ (assuming all lags equally likely a priori). These responsibilities are summed across the sequence and fed into a softmax function to produce the estimate $\hat{\boldsymbol{\lambda}}^{MD}$ with the learning rate $\eta$ controlling its sharpness.

## 4   TRANSFORMERS IMPLEMENT ONE-STEP OF MIRROR DESCENT

We present our main theoretical result: a constructive proof that the one-step MD estimator can be implemented by a Transformer. The crucial mechanism relies on relative position encodings to correctly route information, allowing the self-attention layer to compute the posterior responsibilities.

**Proposition 3** (Transformer Implementation of the One-Step MD Estimator)**.** *Given an MTD model of order $m$ with a known transition matrix $\boldsymbol{\pi}^\star \in \mathcal{P}$, for any sequence $\boldsymbol{y}_{1:T}$ of length $T \geq m$, there exists a three-layer disentangled Transformer $\tilde{\mathcal{T}}$ with single head, $d_0 = q$ and $d_R \geq m$ that implements the one-step MD estimator. The Transformer produces at position $T$ the predictive distribution for token $Y_T$ given context $\boldsymbol{y}_1^{T-1}$:*

$$
\tilde{\mathcal{T}}(\boldsymbol{y}_{1:T})_T = \sum_{g=1}^{m} \tilde{\lambda}_g(\boldsymbol{y}_{1:T}) \cdot \boldsymbol{\pi}^\star(y_{T-g}, :) \qquad \tilde{\lambda}_g(\boldsymbol{y}_{1:T}) = \frac{\exp\left(\frac{\beta}{T-m} \sum_{i=m+1}^{T} \gamma_i(g)\right)}{\sum_{h=1}^{m} \exp\left(\frac{\beta}{T-m} \sum_{i=m+1}^{T} \gamma_i(h)\right)}, \quad (7)
$$

*with the weights $\tilde{\boldsymbol{\lambda}}(\boldsymbol{y}_{1:T})$ computed exactly as the one-step MD estimate and $\beta$ is a learnable parameter corresponding to the scaled learning rate of the MD algorithm.*

In the following we prove Proposition 3 by explicitly constructing a 3-layer disentangled Transformer that implements the one-step MD estimator. The first layer computes the posterior responsibilities $\gamma_i(g)$, the second layer computes the logits $\sum_{i=m+1}^{T} \gamma_i(g)$, and the third layer produces the final estimate vector $\tilde{\boldsymbol{\lambda}}(\boldsymbol{y}_{1:T})$ within its attention weights[1].

**Layer 1, Posterior Responsibilities:** The first layer uses the attention matrix $\boldsymbol{W}_A^{(1)}$ to compute the posterior responsibilities and the relative positional encoding $r'_{ij}$ to store it in the residual stream:

$$
\boldsymbol{W}_A^{(1)} := (\log \boldsymbol{\pi}^\star)^\top, \quad (\boldsymbol{R}_A^{(1)})_{k,:} = \begin{cases} +\delta_1 \cdot \boldsymbol{1}^\top & 2 \leq k \leq m+1 \\ -\delta_1 \cdot \boldsymbol{1}^\top & \text{otherwise} \end{cases}, \quad (\boldsymbol{R}_V^{(1)})_{k,:} = \begin{cases} \boldsymbol{e}_{k-1}^\top & 2 \leq k \leq m+1 \\ \boldsymbol{0}^\top & \text{otherwise} \end{cases}
$$

$$
\boldsymbol{R}_A^{(1)\top} = \delta_1 \begin{pmatrix} -1 & 1 & \cdots & 1 & -1 & \cdots & -1 \\ -1 & 1 & \cdots & 1 & -1 & \cdots & -1 \\ \vdots & \vdots & \ddots & \vdots & \vdots & & \vdots \\ -1 & 1 & \cdots & 1 & -1 & \cdots & -1 \end{pmatrix} \qquad \boldsymbol{R}_V^{(1)\top} = \begin{pmatrix} 0 & 1 & 0 & \cdots & 0 & 0 & \cdots & 0 \\ 0 & 0 & 1 & \cdots & 0 & 0 & \cdots & 0 \\ \vdots & \vdots & \ddots & \vdots & \vdots & \vdots & & \vdots \\ 0 & 0 & \cdots & 0 & 1 & 0 & \cdots & 0 \end{pmatrix}
$$

where $\log$ is applied element-wise, $k = i - j + 1$ is the relative position between token $i$ and $j$ shifted by $1$. The lookup table $\boldsymbol{R}_A^{(1)}$ biases attention to focus only on the first $m$ relative positions (the lags). The lookup table $\boldsymbol{R}_V^{(1)}$ uses one-hot vectors $\boldsymbol{e}_k$ to copy the computed attention weight for a specific lag into the corresponding dimension of the output vector, thereby storing the responsibilities for

---

[1]The attention-based computation of responsibilities in the first layer is shared with the construction of D'Angelo et al. (2025). However, our aggregation and prediction layers leverage RPE value embeddings to store responsibilities in dedicated dimensions, yielding a single-head construction.

different lags in distinct positions. The attention score $e_{ij}$ for this layer is computed as per Equation 1. Given that the input is one-hot encoded, i.e., $\boldsymbol{h}_i^{(0)} = \boldsymbol{e}_{y_i}$, the score becomes:

$$e_{ij} = \boldsymbol{e}_{y_i}^\top (\log \boldsymbol{\pi}^\star)^\top \boldsymbol{e}_{y_j} + \boldsymbol{e}_{y_i}^\top \boldsymbol{r}_{ij}^{(1)} = \log \pi^\star(y_j, y_i) + \begin{cases} +\delta_1 & \text{if } 1 \le i - j \le m \\ -\delta_1 & \text{otherwise} \end{cases},$$

where we used that the RPE vector $\boldsymbol{r}_{ij}$ is constant with respect to the token values $y_i$. For a large $\delta_1$, the softmax only attends to keys $j$ such that their relative position $k = i - j$ is within the range $[1, m]$. The causal attention weights $\mathcal{A}_{ij}^{(1)}$ for $j \le i$ are given by: $\mathcal{A}_{ij}^{(1)} = \frac{\exp(e_{ij})}{\sum_{j'=1}^i \exp(e_{ij'})} = \frac{\pi^\star(y_j, y_i) \exp(\boldsymbol{e}_{y_i}^\top \boldsymbol{r}_{ij}^{(1)})}{\sum_{j'=1}^i \pi^\star(y_{j'}, y_i) \exp(\boldsymbol{e}_{y_i}^\top \boldsymbol{r}_{ij'}^{(1)})}$. In the limit $\delta_1 \to \infty$, the behavior of the softmax changes based on the position $i$. For the first token ($i = 1$), the only valid key is itself ($j' = 1$), which means the attention is entirely self-contained, resulting in $\mathcal{A}_{11}^{(1)} = 1$. For any subsequent token ($i > 1$), the terms in the denominator corresponding to lags $k \in [1, m]$ are scaled by $\exp(\delta_1)$, while all other terms, including the diagonal ($k = 0$), are scaled by $\exp(-\delta_1)$ and vanish. This yields the following limit for $i > 1$:

$$\mathcal{A}_{ij}^{(1)} = \begin{cases} \dfrac{\pi^\star(y_j, y_i)}{\sum\limits_{k=1}^{\hat{m}} \pi^\star(y_{i-k}, y_i)} & i - j \in [m] \\ 0 & \text{otherwise} \end{cases} \qquad \mathcal{A}^{(1)} = \begin{pmatrix} 1 & 0 & 0 & 0 & 0 & 0 & 0 & \cdots \\ * & 0 & 0 & 0 & 0 & 0 & 0 & \cdots \\ * & * & 0 & 0 & 0 & 0 & 0 & \cdots \\ \vdots & \vdots & \vdots & \ddots & \vdots & \vdots & \vdots & \vdots \\ 0 & \cdots & \gamma_{T-1}(m) & \cdots & \gamma_{T-1}(2) & \gamma_{T-1}(1) & 0 & \cdots \\ 0 & \cdots & 0 & \gamma_T(m) & \cdots & \gamma_T(2) & \gamma_T(1) & 0 \end{pmatrix}$$

where $\hat{m} = \min(i - 1, m)$. For positions $i > m$, where the MTD model is defined, the attention mechanism thus computes $\mathcal{A}_{ij}^{(1)} = \gamma_i(i - j)$ for the active lags ($1 \le i - j \le m$). This expression is precisely the posterior responsibility $\gamma_i(g)$ from Equation 6, where the lag is $g = i - j$. For earlier steps ($1 < i \le m$), the attention mechanism computes the natural counterpart by normalizing over the $i - 1$ available lags. The resulting attention matrix $\mathcal{A}^{(1)}$ has a specific banded lower-triangular structure. The output of this layer, $\hat{\boldsymbol{h}}_i^{(1)}$, is the attention-weighted sum over the concatenated value vectors. Given that the attention weights $\mathcal{A}_{ij}$ compute the posterior responsibilities $\gamma_i(g)$ (for $i > m$), and $\boldsymbol{R}_V^{(1)}$ embeds the lag $g = i - j$ as a one-hot vector $\boldsymbol{e}_g$, the output for a position $i > m$ is:

$$\hat{\boldsymbol{h}}_i^{(1)} = \sum_{j=1}^{i-1} \mathcal{A}_{ij} \, \text{Concat}(\boldsymbol{e}_{y_j}, \boldsymbol{r}_{ij}'^{(1)}) = \sum_{g=1}^m \gamma_i(g) \, \text{Concat}(\boldsymbol{e}_{y_{i-g}}, \boldsymbol{e}_g). \tag{8}$$

This output is a convex combination, weighted by the posterior responsibilities, of vectors that each concatenate two pieces of information: the one-hot encoding of the token at a given lag ($\boldsymbol{e}_{y_{i-g}}$) and the one-hot encoding of $\boldsymbol{e}_g$ of the lag itself ($g$):

$$\hat{\boldsymbol{h}}_i^{(1)} = \underbrace{\gamma_i(m) \begin{pmatrix} \boldsymbol{e}_{y_{i-m}} \\ 0 \\ \vdots \\ 1 \end{pmatrix}}_{\text{Lag } m} + \cdots + \underbrace{\gamma_i(2) \begin{pmatrix} \boldsymbol{e}_{y_{i-2}} \\ 0 \\ 1 \\ \vdots \end{pmatrix}}_{\text{Lag } 2} + \underbrace{\gamma_i(1) \begin{pmatrix} \boldsymbol{e}_{y_{i-1}} \\ 1 \\ 0 \\ \vdots \end{pmatrix}}_{\text{Lag } 1} = \begin{pmatrix} \sum_{g=1}^m \gamma_i(g) \boldsymbol{e}_{y_{i-g}} \\ \gamma_i(1) \\ \gamma_i(2) \\ \vdots \\ \gamma_i(m) \end{pmatrix}$$

In essence, the top part of $\hat{\boldsymbol{h}}_i^{(1)}$ contains a weighted sum of past tokens (which will not be used), while its bottom part explicitly stores the vector of posterior responsibilities with the value $\gamma_i(g)$ for lag $g$ stored at the $g$-th position within this second block (i.e., $\gamma_i(1)$ is first, followed by $\gamma_i(2)$, etc.).

**Layer 2, Summing Responsibilities:** The second layer sums along the sequence the responsibility vectors computed in Layer 1, for tokens at positions $i > m$. This is achieved by setting the content-based attention to zero ($\boldsymbol{W}_A^{(2)} = \boldsymbol{0}$) and the value-rpe matrix to zero ($\boldsymbol{R}_V^{(2)} = \boldsymbol{0}$). The mechanism relies on the content-position interaction $(\boldsymbol{h}_i^{(1)})^\top \boldsymbol{r}_{ij}^{(2)}$. Crucially, the input vector $\boldsymbol{h}_i^{(1)} = \text{Concat}(\boldsymbol{e}_{y_i}, \hat{\boldsymbol{h}}_i^{(1)})$ retains in the residual stream the one-hot embedding $\boldsymbol{e}_{y_i}$ of the current token in its first $q$ dimensions. The RPE table $\boldsymbol{R}_A^{(2)}$ is structured to interact only with this part of the vector, turning the dot product into a fixed bias:

$$\boldsymbol{W}_A^{(2)} = \boldsymbol{0}_{d_1 \times d_1}, \quad (\boldsymbol{R}_A^{(2)})_{k,:} = \begin{cases} \boldsymbol{0}_{1 \times d_1} & \text{if } 1 \le k \le T - m \\ [-\delta_2 \cdot \boldsymbol{1}_{1 \times q}, \boldsymbol{0}_{1 \times (q+m)}] & \text{otherwise} \end{cases}, \quad \boldsymbol{R}_V^{(2)} = \boldsymbol{0}_{T \times m}$$

The attention score for the final query at $i = T$ therefore simplifies to:

$$e_{Tj} = (\boldsymbol{h}_T^{(1)})^\top \boldsymbol{r}_{Tj}^{(2)} = \begin{cases} (\boldsymbol{h}_T^{(1)})^\top \begin{pmatrix} -\delta_2 \cdot \mathbf{1}_q \\ \mathbf{0}_{q+m} \end{pmatrix} = -\delta_2 \cdot (\boldsymbol{e}_{y_T}^\top \mathbf{1}_q) = -\delta_2 & \text{if } 1 \le j \le m \\ (\boldsymbol{h}_T^{(1)})^\top \mathbf{0}_{d_1} = 0 & \text{otherwise} \end{cases}$$

In the limit $\delta_2 \to \infty$, the softmax places uniform attention only on the keys where the score is not $-\infty$. This results in uniform attention weights $\mathcal{A}_{Tj} = 1/(T - m)$ for $j \in [m + 1, T]$, and zero otherwise. The layer's output for the final token is therefore the exact average of the desired vectors:

$$\hat{\boldsymbol{h}}_T^{(2)} = \sum_{j=1}^{T} \mathcal{A}_{Tj} \, \mathrm{Concat}(\boldsymbol{h}_j^{(1)}, \boldsymbol{r}'_{Tj}) = \frac{1}{T - m} \sum_{j=m+1}^{T} \mathrm{Concat}(\boldsymbol{h}_j^{(1)}, \mathbf{0}) = \frac{1}{T - m} \begin{pmatrix} \sum_{j=m+1}^{T} \boldsymbol{e}_{y_j} \\ \sum_{j=m+1}^{T} \left( \sum_{g=1}^{m} \gamma_j(g) \boldsymbol{e}_{y_{j-g}} \right) \\ \hline \sum_{j=m+1}^{T} \gamma_j(1) \\ \sum_{j=m+1}^{T} \gamma_j(2) \\ \vdots \\ \sum_{j=m+1}^{T} \gamma_j(m) \\ \hline \mathbf{0} \end{pmatrix}$$

Defining $\boldsymbol{\Gamma}_j = (\gamma_j(1), \gamma_j(2), \ldots, \gamma_j(m))^\top$, the $m$-dimensional sub-block at positions $2q+1$ to $2q+m$ of $\hat{\boldsymbol{h}}_T^{(2)}$ contains the averaged responsibility vector $\frac{1}{T-m} \sum_{j=m+1}^{T} \boldsymbol{\Gamma}_j$, which is exactly the quantity needed to implement the one-step MD estimator in Prop. 2.

**Layer 3, Final Predictive Weights:** The third and final layer uses the averaged responsibilities computed in Layer 2 to produce the final predictive weights, $\tilde{\boldsymbol{\lambda}}$, which correspond to the one-step MD estimate from Prop. 2. This is accomplished by using the RPE table, $\boldsymbol{R}_A^{(3)}$, to perform a selective dot product. The query vector for the final token, $\boldsymbol{h}_T^{(2)}$, contains the vector of averaged responsibilities in the $\boldsymbol{\Gamma}$ sub-block of $\hat{\boldsymbol{h}}_T^{(2)}$. The RPE vectors in $\boldsymbol{R}_A^{(3)}$ are constructed as scaled one-hot vectors that align with this sub-block, effectively using the dot product to "read out" the corresponding averaged responsibility. Similarly to layer 2, to only have non-zero attention at the positions corresponding to the $m$ lags, we use the one-hot embedding of the current token $\boldsymbol{e}_{y_i}$ from the residual stream $\boldsymbol{h}_i^{(2)} = \mathrm{Concat}(\boldsymbol{e}_{y_i}, \hat{\boldsymbol{h}}_i^{(1)}, \hat{\boldsymbol{h}}_i^{(2)})$ to add a fixed bias $\delta_3$ which drives the softmax to zero in the limit. The content-based attention and value-rpe are again disabled:

$$\boldsymbol{W}_A^{(3)} = \mathbf{0}_{d_2 \times d_2}, \quad (\boldsymbol{R}_A^{(3)})_{k,:} = \begin{cases} [\overbrace{+\delta_3 \cdot \mathbf{1}_q^\top}^{\text{on } \boldsymbol{h}^{(0)}}, \overbrace{\mathbf{0}_{q+m}^\top}^{\text{on } \hat{\boldsymbol{h}}^{(1)}}, \overbrace{\mathbf{0}_{2q}^\top}^{\text{on } \hat{\boldsymbol{h}}_{1:2q}^{(2)}}, \overbrace{\beta \cdot \boldsymbol{e}_k^\top}^{\text{on } \boldsymbol{\Gamma}}, \overbrace{\mathbf{0}_m^\top}^{\text{on } \hat{\boldsymbol{h}}_{2q+m+1:2q+2m}^{(2)}} ] & \text{if } k \in [m] \\ [-\delta_3 \cdot \mathbf{1}_q^\top, \mathbf{0}_{q+m}^\top, \mathbf{0}_{2q}^\top, \mathbf{0}_m^\top, \mathbf{0}_m^\top] & \text{otherwise} \end{cases}$$

$$\boldsymbol{R}_V^{(3)} = \mathbf{0}_{T \times m}$$

Here, $\boldsymbol{e}_k$ is a one-hot vector in $\mathbb{R}^m$ that selects the coordinate corresponding to the $k$-th responsibility in the $\boldsymbol{\Gamma}$ sub-block of $\boldsymbol{h}_T^{(2)}$, and $\beta$ is the learnable scaled learning rate. Visually, the RPE table $\boldsymbol{R}_A^{(3)}$ is a sparse matrix of scaled one-hot vectors:

$$\boldsymbol{R}_A^{(3)\top} = \begin{matrix} \boldsymbol{h}^{(0)} \\ \hat{\boldsymbol{h}}^{(1)} \\ \hat{\boldsymbol{h}}_{1:2q}^{(2)} \\ \boldsymbol{\Gamma} \\ \hat{\boldsymbol{h}}_{2q+m+1:2q+2m}^{(2)} \end{matrix} \begin{pmatrix} \begin{matrix} [1, \cdots, m] \\ +\delta_3 \cdot \mathbf{1}_q, \cdots, +\delta_3 \cdot \mathbf{1}_q \\ 0 \\ 0 \\ [\beta\boldsymbol{e}_1 \; \beta\boldsymbol{e}_2 \; \cdots \; \beta\boldsymbol{e}_m] \\ 0 \end{matrix} & \begin{matrix} \cdots \\ -\delta_3 \cdot \mathbf{1}_q \\ 0 \\ 0 \\ 0 \\ 0 \end{matrix} & \begin{matrix} \cdots \\ -\delta_3 \cdot \mathbf{1}_q \\ 0 \\ 0 \\ 0 \\ 0 \end{matrix} & \begin{matrix} T \\ -\delta_3 \cdot \mathbf{1}_q \\ 0 \\ 0 \\ 0 \\ 0 \end{matrix} \end{pmatrix}$$

Since $\boldsymbol{W}_A^{(3)} = \mathbf{0}_{d_2 \times d_2}$, the score simplifies to the content-position interaction $(\boldsymbol{h}_T^{(2)})^\top \boldsymbol{r}_{Tj}^{(3)}$ and $\boldsymbol{r}_{Tj}^{(3)}$ is constructed to be non-zero only for relative positions $g = T - j + 1 \in [1, m]$. For these lags, the RPE is a scaled one-hot vector that acts as a selector, using the dot product to extract the corresponding averaged responsibility stored in $\boldsymbol{h}_T^{(2)}$:

$$e_{Tj} = (\boldsymbol{h}_T^{(2)})^\top \boldsymbol{r}_{Tj}^{(3)} = \begin{cases} (\boldsymbol{h}_T^{(0)})^\top (\delta_3 \mathbf{1}_q) + (\boldsymbol{\Gamma})_T^\top (\beta \boldsymbol{e}_g) \\ -\delta_3 \cdot (\boldsymbol{e}_{y_T}^\top \mathbf{1}_q) \end{cases} = \begin{cases} +\delta_3 + \beta \cdot \frac{\sum_{i=m+1}^{T} \gamma_i(g)}{T - m} & \text{if } g \in [m] \\ -\delta_3 & \text{otherwise.} \end{cases}$$

The attention scores for the final token, $e_{T,:}$, becomes the scaled averaged responsibility:

$$\mathbf{e}_{T,:} = \Big( -\delta_3, \ldots, -\delta_3, \underbrace{+\delta_3 + \beta \frac{\sum_{i=m+1}^{T} \gamma_i(m)}{T-m}}_{\text{pos } T-m+1}, \ldots, \underbrace{+\delta_3 + \beta \frac{\sum_{i=m+1}^{T} \gamma_i(2)}{T-m}}_{\text{pos } T-1}, \underbrace{+\delta_3 + \beta \frac{\sum_{i=m+1}^{T} \gamma_i(1)}{T-m}}_{\text{pos } T} \Big)$$

After applying the softmax and in the limit of large $\delta_3$, the attention weights for the last token compute the MD estimate of the mixture weights $\tilde{\lambda}_g$ and place them at the correct positions:

$$\lim_{\delta_3 \to \infty} \mathcal{A}_{T,T-g} = \frac{\exp\left(\beta \frac{\sum_{i=m+1}^{T} \gamma_i(g)}{T-m}\right)}{\sum_{k=1}^{m} \exp\left(\beta \frac{\sum_{i=m+1}^{T} \gamma_i(k)}{T-m}\right)} \quad \lim_{\delta_3 \to \infty} \mathcal{A}_{T,:} = \left(\ldots, 0, \underbrace{\tilde{\lambda}_m}_{T-m+1}, \ldots, \underbrace{\tilde{\lambda}_2}_{T-1}, \underbrace{\tilde{\lambda}_1}_{T}\right).$$

This final attention operation is the core of the estimation process. It computes the mixture weights $\tilde{\boldsymbol{\lambda}}$, which serve as the in-context estimates of token importance, thus realizing the mechanism of in-context learning this work seeks to understand.

**Output Layer:** The final step of the construction is to apply the output matrix $\widetilde{\boldsymbol{W}}_O$ to the final hidden state $\boldsymbol{h}_T^{(3)}$ to produce the predictive distribution over the next token. The matrix $\widetilde{\boldsymbol{W}}_O$ learns the known transition matrix $\boldsymbol{\pi}^\star$ and selectively applies it to the embedding of the last token. The output of the third attention layer, $\hat{\boldsymbol{h}}_T^{(3)}$, is a weighted sum of the hidden states from the second layer, where the weights are the estimated mixture weights $\tilde{\lambda}_g$: $\hat{\boldsymbol{h}}_T^{(3)} = \sum_{j=1}^{T} \mathcal{A}_{Tj}^{(3)} \boldsymbol{h}_j^{(2)} = \sum_{g=1}^{m} \tilde{\lambda}_g \boldsymbol{h}_{T-g}^{(2)}$ (where we drop the zero value-RPE block since $\boldsymbol{R}_V^{(3)} = \boldsymbol{0}_{T \times m}$). The first $q$ components of any hidden state $\boldsymbol{h}_j^{(k)}$, due to the residual stream, simply contain the original input $\boldsymbol{h}_j^{(0)} = \boldsymbol{e}_{y_j}$. Consequently, the first $q$ components of $\hat{\boldsymbol{h}}_T^{(3)}$ are a $\tilde{\lambda}$-weighted combination of the one-hot embeddings of the relevant past tokens: $(\hat{\boldsymbol{h}}_T^{(3)})_{1:q} = \sum_{g=1}^{m} \tilde{\lambda}_g (\boldsymbol{h}_{T-g}^{(2)})_{1:q} = \sum_{g=1}^{m} \tilde{\lambda}_g \boldsymbol{e}_{y_{T-g}}$. The full hidden state is $\boldsymbol{h}_T^{(3)} = \text{Concat}(\boldsymbol{h}_T^{(0)}, \hat{\boldsymbol{h}}_T^{(1)}, \hat{\boldsymbol{h}}_T^{(2)}, \hat{\boldsymbol{h}}_T^{(3)})$ and the output matrix $\widetilde{\boldsymbol{W}}_O \in \mathbb{R}^{q \times d_3}$ is structured to ignore all preceding blocks and operate only on the first $q$ components of the final block, $\hat{\boldsymbol{h}}_T^{(3)}$. This is achieved by storing the transition matrix, $\boldsymbol{\pi}^{\star\top}$, in the corresponding sub-block:

$$\widetilde{\boldsymbol{W}}_O = (\overset{\text{from } \boldsymbol{h}_T^{(0)}}{\boldsymbol{0}_{q \times q}} \mid \overset{\text{from } \hat{\boldsymbol{h}}_T^{(1)}}{\boldsymbol{0}_{q \times (q+m)}} \mid \overset{\text{from } \hat{\boldsymbol{h}}_T^{(2)}}{\boldsymbol{0}_{q \times (2q+2m)}} \mid \overset{\text{from } \hat{\boldsymbol{h}}_T^{(3)}}{[\boldsymbol{\pi}_{q \times q}^{\star\top} \quad \boldsymbol{0}_{q \times (3q+3m)}]}) .$$

Applying this matrix to the fully expanded final hidden state yields the predictive distribution:

$$\tilde{\mathcal{T}}(\boldsymbol{y}_{1:T})_T = [\boldsymbol{\pi}^{\star\top} \quad \boldsymbol{0}] \, \hat{\boldsymbol{h}}_T^{(3)} = \boldsymbol{\pi}^{\star\top} (\hat{\boldsymbol{h}}_T^{(3)})_{1:q} = \boldsymbol{\pi}^{\star\top} \left(\sum_{g=1}^{m} \tilde{\lambda}_g \boldsymbol{e}_{y_{T-g}}\right) = \sum_{g=1}^{m} \tilde{\lambda}_g \boldsymbol{\pi}^\star(y_{T-g}, :) .$$

This final vector is exactly the predictive distribution from Proposition 3, completing the proof.

## 5 WHY ONE-STEP MIRROR DESCENT WORKS

We now turn to a theoretical analysis of the one-step MD estimator. We prove that a single mirror descent update, initialized at the uniform prior, recovers a first-order approximation of the Bayesian posterior mean. This result provides a formal justification for the effectiveness of the non-iterative estimator implemented by our construction.

**One-Step MD as a First-Order Bayesian Approximation:** We establish a theoretical connection between the one-step Mirror Descent (MD) estimator and the Bayesian posterior mean. We show that their first-order Taylor expansions around the state of no evidence coincide up to a scalar constant. This result justifies interpreting the one-step MD estimator as a principled approximation to the Bayes-optimal predictor, especially in low-data regimes. The analysis hinges on treating both estimators as functions of the log-likelihood gradient evaluated at the center of the simplex, $\boldsymbol{g} \coloneqq \nabla_{\boldsymbol{\lambda}} \ell(\boldsymbol{\lambda}^{(0)})$, and expanding them around the point of no evidence, $\boldsymbol{g} = \boldsymbol{0}$.

**Theorem 1** (First-Order Equivalence of the Estimators). *Let $\hat{\boldsymbol{\lambda}}^{MD}(\boldsymbol{g}; \eta)$ be the one-step MD estimator with learning rate $\eta$, and let $\hat{\boldsymbol{\lambda}}^{Bayes}(\boldsymbol{g})$ be the Bayesian posterior mean under the linearized likelihood. The two estimators are first-order equivalent at $\boldsymbol{g} = \boldsymbol{0}$ for $\eta = \frac{1}{m+1}$.*

**Learning-rate scaling via a Lipschitz (smoothness) constant:** We established a first-order equivalence between the one-step MD estimator and the Bayesian posterior mean at $\boldsymbol{g} = \boldsymbol{0}$ (the "no-evidence" regime). For a sequence of length $T$, however, the gradient norm $|\boldsymbol{g}|$ scales with $T$ (see

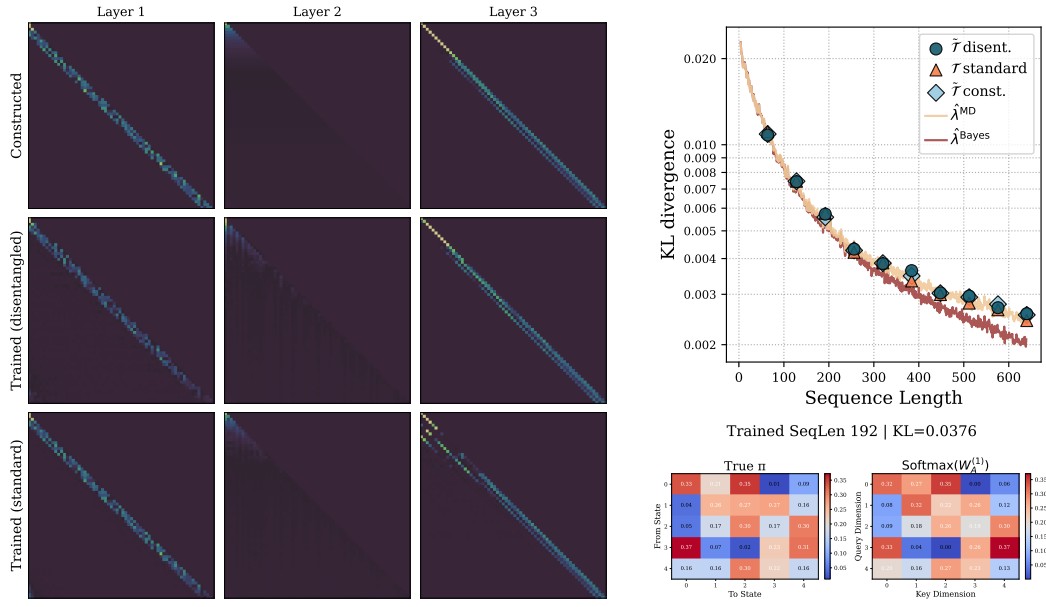

Figure 4: **Comparison of Trained and Constructed Transformers. Left:** Attention maps of the trained transformer (disentangled and standard) versus our theoretical construction (seq. length 64). **Right (top):** KL divergence to the ground truth transition probabilities for the trained transformers, the constructed transformer, and the one-step MD estimator across sequence lengths. **Right (bottom):** First-layer attention softmax Softmax$(\boldsymbol{W}_1^{A\top})$ vs. true transitions matrix $\pi^\star$ for a trained model.

App. J), raising the question of how to scale the learning rate of $\hat{\boldsymbol{\lambda}}^{\mathrm{MD}} = \mathrm{softmax}(\eta\boldsymbol{g})$ with $T$. Mirror Descent theory suggests choosing the learning rate inversely proportional to the relative smoothness constant $L_{\mathrm{rel}}$ (Bauschke et al., 2017). Because our MD update is derived from the negative entropy potential, smoothness is defined w.r.t. the KL divergence. Convergence requires $\eta \leq 1/L_{\mathrm{rel}}$. We now bound this constant and find exactly the scaling implemented in the Transformer in Prop. 3.

**Theorem 2** (Relative Smoothness and $\eta$ scaling). *At the center of the simplex $\boldsymbol{\lambda} = (1/m, \ldots, 1/m)$, the loss $f(\boldsymbol{\lambda}) = -\ell(\boldsymbol{\lambda})$ is $L_{rel}$-smooth relative to the KL-divergence, with $L_{rel} \leq (T - m)\, m^2$. Consequently the stable step-size rule $\eta \leq \frac{1}{L_{rel}}$ yields the asymptotic scaling $\eta = \Theta\big(\frac{1}{T}\big)$ for fixed $m$.*

**Beyond the Local Approximation, The Implicit Regularization of Mirror Descent:**
The first-order equivalence in Theorem 1 holds only for short sequences, where the log-likelihood gradient is small. For longer sequences, neglected higher-order terms become significant, and the one-step estimator diverges from the Bayesian mean. Empirically, however, a few additional Mirror Descent steps substantially reduce this gap (see Figure 5). In this regime, iterating MD to convergence yields the suboptimal MLE, while early stopping provides a much closer match to the Bayesian mean. We propose that this effect arises from implicit regularization. Specifically, early-stopped MD approximately solves an entropy-regularized optimization problem of the form $\min_\lambda -\ell(\lambda) + \gamma H(\lambda)$ with the iterates tracking the corresponding regularization path (Suggala et al., 2018). Along this path, performance on par with the Bayes-optimal estimator is achieved for an appropriate choice of regularization $\gamma$ (see Figure 3). Thus, early stopping is effectively equivalent to selecting this favorable point on the entropy-regularized path, avoiding the suboptimal MLE.

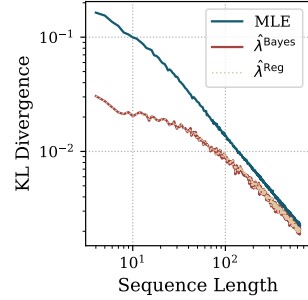

Figure 3: **Regularized Estimator.** Comparison with Bayes and MLE estimators.

## 6 EXPERIMENTS

We validate our main claims below; further details and additional experiments are in App. B and D.
**Setup:** We train 3-layer disentangled transformers $\tilde{\mathcal{T}}_{\mathrm{disent}}$ with single head with learned relative

positional and one-hot semantic embeddings as well as 3-layer standard transformers $\mathcal{T}_{\text{standard}}$ (no disentanglement) with single head attention with standard parameterization given by Query-Key-Value with learned relative positional and learned semantic embeddings. We sample a fixed $q$, $m$, $\boldsymbol{\pi}$ and at each iteration we sample a set of $\{\boldsymbol{\lambda_i}\}_{i=1}^B$ with $B = 128$ the batch size and $\boldsymbol{\lambda_i} \sim \text{Dir}(\boldsymbol{\alpha})$ and generate $B$ sequences according to the MTD model. We train the model for $5 \times 10^5$ iterations using the Adam optimizer and MSE loss over the last token in the sequence for various sequence lengths.

**Results for one-step MD:** We plot the KL divergence with the ground truth transition probabilities between the trained transformer $\tilde{\mathcal{T}}_{\text{disent.}}$ and $\mathcal{T}_{\text{standard.}}$ compared to our theoretical construction $\tilde{\mathcal{T}}_{\text{constr.}}$, the one-step MD estimator $\hat{\lambda}^{\text{MD}}$ and the optimal-Bayes estimator $\hat{\lambda}^{\text{Bayes}}$ (we compute it via MCMC; see App. F.1) for various sequence lengths in Figure 4 (right). We observe that both the disentangled and standard trained transformers match the performance of the theoretical construction and the one-step MD estimator: for small sequence lengths, the latter serves as a good proxy for the optimal Bayes estimator, validating our theoretical result in Theorem 1, whereas for longer sequences it becomes suboptimal. By inspecting the attention maps of the trained transformers in Figure 4 (left) we can see that they learn to extract the responsibilities $\gamma(g)_i$ as expected from our construction (for all three models, the locations of low and high attention entries, and the diagonal structure induced by the MTD order, are closely aligned). To further validate if the attention matrix in the first layer actually learns the ground truth transition matrix $\pi^\star$ we plot the heatmap of the first attention $\text{softmax}(\boldsymbol{W}_1^{A\top})$ vs. $\pi^\star$ as well as the average row-wise KL divergence between the two matrices Figure 4 (bottom-right) more results in App. D. For both the $\hat{\lambda}^{\text{MD}}$ and $\tilde{\mathcal{T}}_{\text{constr.}}$ we tune the learning rate $\beta$ and $\eta$ via grid search to minimize the KL divergence (see Section B.1 for more details).

**Results multi-step MD:** To investigate whether deeper Transformers can learn to implement multiple steps of Mirror Descent, we plot in Figure 5 the KL divergence for a 5-layer trained transformer $\tilde{\mathcal{T}}_{\text{train.}}$ compared to the multi-step MD estimator $\hat{\lambda}^{\text{MD},k=i}$ ($i$ denoting the number of steps) across various sequence lengths, averaged over 3 seeds with error bars. We observe that the trained transformer closely tracks the performance of the 2-step MD estimator. We stress that this is a performance comparison, not a convergence or optimality claim: we do not assert that the transformer converges to the 2-step MD solution. Rather, the experiment suggests that deeper transformers are capable of implementing estimators whose accuracy is at least comparable to that of multi-step MD. Notably, for longer sequences where the gap between the 2-step MD and the Bayes-optimal solution widens, the transformer performance exhibits some variance across seeds, occasion-

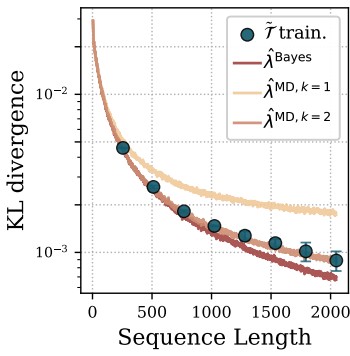

Figure 5: **Multi-Step MD vs. 5-Layer Transformer.**

ally falling below the 2-step curve. This hints that the learned estimator may not be tied to the 2-step MD and could, in principle, exploit additional structure. Extending our explicit construction to the multi-step setting is nontrivial and we leave it to future work; however, the empirical finding that a 5-layer transformer suffices to match 2-step performance suggests that intermediate representations (e.g., the responsibilities computed in earlier layers) may be reused across steps.

## 7 CONCLUSIONS

We set out to understand whether transformers can learn latent causal structures in-context, and what algorithm they implement to do so. To this end, we introduced a framework based on MTD that formalizes token-importance estimation as in-context inference of latent variables. Our central finding is that transformers solve this task by implementing Mirror Descent. We gave an explicit three-layer construction that exactly implements one step of the algorithm and proved that the resulting estimator is a first-order approximation of the Bayes-optimal predictor, with a learning-rate scaling $\eta = \Theta(1/T)$ matching the transformer's learned behavior. Empirically, transformers trained from scratch, both disentangled and standard, recover the constructed solution across predictive distributions and attention patterns, while deeper models achieve performance comparable to multi-step Mirror Descent. Taken together, our results extend the gradient-based interpretation of in-context learning from regression to sequential domains over discrete tokens, offering a new algorithmic lens for understanding latent-variable inference in attention-based models.

ACKNOWLEDGEMENTS

This work was partially funded by an unrestricted gift from Coefficient Giving, and the grant number 212111 from the Swiss National Science Foundation. Francesco D'Angelo is supported by the Google PhD Fellowships.

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

# A NOTATION

Throughout this paper, we use non-bold letters for scalars (e.g., $\eta, \alpha$), lowercase bold letters for vectors (e.g., $\boldsymbol{h}, \boldsymbol{\lambda}$), and uppercase bold letters for matrices (e.g., $\boldsymbol{W}, \boldsymbol{H}, \boldsymbol{\pi}$). The $i$-th element of a vector $\boldsymbol{v}$ is denoted by $v_i$, and the vector at position $i$ in a sequence of vectors $\boldsymbol{H}$ is written as $\boldsymbol{h}_i$. The element in the $i$-th row and $j$-th column of a matrix $\boldsymbol{A}$ is $A_{ij}$, and its $i$-th row vector is $\boldsymbol{A}_{i,:}$. We use $\boldsymbol{1}$ and $\boldsymbol{0}$ to denote vectors or matrices of ones and zeros, respectively, with dimensions inferred from context. We use $\boldsymbol{e}_k$ to denote a one-hot vector with a one at the $k$-th position; its dimensionality is specified or clear from context. The set of integers $\{1, \ldots, m\}$ is denoted by $[m]$. The probability simplex in $\mathbb{R}^m$ is denoted by $\Delta_{m-1}$. The operator $\mathrm{Concat}(\cdot, \cdot)$ denotes the vertical concatenation of vectors or matrices. For vectors $\boldsymbol{a} \in \mathbb{R}^{d_a}$ and $\boldsymbol{b} \in \mathbb{R}^{d_b}$, their concatenation results in a vector in $\mathbb{R}^{d_a+d_b}$. For matrices $\boldsymbol{A} \in \mathbb{R}^{d_A \times T}$ and $\boldsymbol{B} \in \mathbb{R}^{d_B \times T}$ with the same number of columns, $\mathrm{Concat}(\boldsymbol{A}, \boldsymbol{B})$ is the block matrix $\left(\begin{smallmatrix} \boldsymbol{A} \\ \boldsymbol{B} \end{smallmatrix}\right) \in \mathbb{R}^{(d_A+d_B) \times T}$. Superscripts in parentheses, such as $\boldsymbol{H}^{(l)}$, are used to index the layers of the Transformer. In the context of Transformer relative positional encodings, we use 1-based indexing for token positions $i, j \in [T]$. The relative position is mapped to a lookup table index $k = i - j + 1$. The approximations will be expressed using Landau notation, where a vector function $\boldsymbol{f}(\boldsymbol{g}) = O(\|\boldsymbol{g}\|^p)$ signifies that $\|\boldsymbol{f}(\boldsymbol{g})\| \leq C\|\boldsymbol{g}\|^p$ for some constant $C$ in a neighborhood of $\boldsymbol{g} = \boldsymbol{0}$.

## A.1 DIMENSIONALITY OF THE TRANSFORMER CONSTRUCTION

The disentangled Transformer architecture results in a hidden state that grows with each layer. The following table provides a summary of the dimensions at each stage of the construction. Note that the input to layer $l$ is $\boldsymbol{h}^{(l-1)}$, and its output is $\boldsymbol{h}^{(l)}$, which is formed by concatenating the input with the result of the attention mechanism, $\hat{\boldsymbol{h}}^{(l)}$. We use $q$ for the alphabet size and $m$ for the MTD model order. The RPE value dimension is $d_R = m$ throughout the construction. Layer 1 uses the value RPE to store lag indices, while Layers 2 and 3 set $\boldsymbol{R}_V^{(l)} = \boldsymbol{0}$, so the concatenation appends $m$ zero dimensions.

Table 1: Dimensionality of Hidden States in the Disentangled Transformer

| Layer | Description | Attention Output $\hat{\boldsymbol{h}}^{(l)}$ | Concatenated Hidden State $\boldsymbol{h}^{(l)}$ |
|---|---|---|---|
| 0 (Input) | Token Embeddings | — | $\boldsymbol{h}^{(0)} \in \mathbb{R}^q$ |
| 1 | Responsibilities | $\hat{\boldsymbol{h}}^{(1)} \in \mathbb{R}^{q+m}$ | $\boldsymbol{h}^{(1)} = \mathrm{Concat}(\boldsymbol{h}^{(0)}, \hat{\boldsymbol{h}}^{(1)}) \in \mathbb{R}^{2q+m}$ |
| 2 | Summation | $\hat{\boldsymbol{h}}^{(2)} \in \mathbb{R}^{2q+2m}$ | $\boldsymbol{h}^{(2)} = \mathrm{Concat}(\boldsymbol{h}^{(1)}, \hat{\boldsymbol{h}}^{(2)}) \in \mathbb{R}^{4q+3m}$ |
| 3 | Weighting | $\hat{\boldsymbol{h}}^{(3)} \in \mathbb{R}^{4q+3m}$ | $\boldsymbol{h}^{(3)} = \mathrm{Concat}(\boldsymbol{h}^{(2)}, \hat{\boldsymbol{h}}^{(3)}) \in \mathbb{R}^{8q+6m}$ |
| **Final Output Matrix** | $\widetilde{\boldsymbol{W}}_O \in \mathbb{R}^{q \times (8q+6m)}$ | | |

# B EXPERIMENTAL DETAILS

This section provides additional details on the experimental setup, including the hyperparameter settings for all models and estimators used in our empirical validation.

## B.1 HYPERPARAMETER TUNING

For the one-step Mirror Descent estimator ($\hat{\lambda}^{\mathrm{MD}}$) and our theoretical Transformer construction ($\widetilde{\mathcal{T}}_{\mathrm{constr.}}$), the learning rate parameters $\eta$ and $\beta$ were not fixed but were tuned to optimize performance. For each sequence length evaluated, we performed a grid search over a range of potential values for $\eta$ and $\beta$. The value that minimized the KL divergence to the true Bayesian posterior mean was selected for the final comparison plots. This ensures that both methods were evaluated under their optimal conditions.

## B.2 PARAMETER SUMMARY

The following table summarizes the key parameters used in our experiments.

Table 2: Summary of Experimental Parameters

| Component | Parameter | Value |
|---|---|---|
| **Data Generation** | | |
| | MTD model order ($m$) | 3,4,5 |
| | Vocabulary size ($q$) | 5 |
| | Sequence Length ($T$) | Varied (64 to 1984) |
| | Dirichlet Prior ($\alpha$) | Uniform ($\alpha_g = 1$ for all $g$) |
| | Transition Matrix ($\pi$) | Rows from Dirichlet ($\alpha = 1$) |
| **Mirror Descent (MD) Estimator** | | |
| | Learning Rate ($\eta$) | Log grid: $[10^{-5}, 10^{-1}]$, 1000 points |
| **Constructed Transformer ($\tilde{\mathcal{T}}_{\mathbf{constr.}}$)** | | |
| | Large Constant ($\delta_1, \delta_2, \delta_3$) | [100,100,100] |
| | Scaled Learning Rate ($\beta$) | Log grid: $[10^{-5}, 10^{-1}]$, 1000 points |
| **Trained Transformer ($\tilde{\mathcal{T}}_{\mathbf{disent.}}$)** | | |
| | Architecture | 3-layer & 5-layer Disentangled Transformer |
| | Attention Heads | 1 |
| | Concatenation | True |
| | Semantic Embeddings | one-hot |
| | Relative Positional Encodings | Learned |
| | RPE value dimension | $m$ |
| | Embedding Dimension | $q$ |
| | QK parametrization | False |
| | Value matrix | False |
| | Head output projection matrix | False |
| | Optimizer | Adam |
| | Learning Rate | $1 \times 10^{-3}$ |
| | LR Schedule | constant |
| | Batch Size | 128 |
| | Training Iterations | $5 \times 10^5$ |
| | Loss Function | MSE on the last token prediction |
| **Trained Transformer ($\mathcal{T}_{\mathbf{standard.}}$)** | | |
| | Architecture | 3-layer Standard Transformer |
| | Attention Heads | 1 |
| | Concatenation | False |
| | Semantic Embeddings | Learned |
| | Relative Positional Encodings | Learned |
| | Embedding Dimension | 32 |
| | QK parametrization | True |
| | Value matrix | True |
| | Head output projection matrix | Tru e |
| | Optimizer | Adam |
| | Learning Rate | $1 \times 10^{-3}$ |
| | LR Schedule | constant |
| | Batch Size | 128 |
| | Training Iterations | $5 \times 10^5$ |
| | Loss Function | MSE on the last token prediction |
| **MCMC for Bayes Estimator** | | |
| | Sampler | Gibbs Sampling |
| | Burn-in Iterations | [200] |
| | Number of Samples (K) | [2000] |

## C  RELATED WORKS

**Induction heads and interpretability** The emergence of ICL, as well as the more general ability of transformers to implement algorithms, has been linked to the formation of interpretable computational circuits Elhage et al. (2021) such as induction heads (Olsson et al., 2022), which are woven into sparse attention patterns (Zucchet et al., 2025). The development of these circuits is not monolithic; rather, they emerge in phases through the interaction of simpler subcircuits. Singh et al. (2024), for instance, use a causal framework to identify the key subcircuits whose interplay leads to the sudden formation of induction heads during training. This emergence itself also critically depends on the training data; specific distributional properties, such as burstiness and class imbalance, have been shown to be key drivers of this capability Chan et al. (2022); Zucchet et al. (2025). While much of this understanding comes from reverse-engineering circuits in pretrained models Conmy et al. (2023), a parallel line of research aims to create transformers that are interpretable by design, for example by training models that can be directly decompiled into human-readable programs Friedman et al. (2023).

**In-Context learning and gradient descent** Following initial empirical observations of ICL in transformers (Brown et al., 2020), a significant line of research has sought to understand its underlying mechanisms. Early work demonstrated that transformers can learn simple function classes like linear models in-context (Garg et al., 2022). This led to the hypothesis that transformer layers effectively implement optimization algorithms, with several studies showing they can perform computations analogous to gradient descent for in-context linear regression (Akyürek et al., 2022; Bai et al., 2023b; Von Oswald et al., 2023a;b). This gradient-based view has been extended to higher-order algorithms (Ahn et al., 2023; Fu et al., 2024) and given theoretical grounding, with proofs that gradient flow converges to a transformer that has learned the in-context task (Zhang et al., 2023). From a statistical learning perspective, this process has been formalized as "algorithm learning", where generalization is guaranteed by the algorithmic stability of the learned procedure (Li et al., 2023). From a Bayesian standpoint, Zhang et al. (2024) formalize ICL as Bayesian model averaging over latent concepts, providing generalization guarantees that complement the algorithmic view. Crucially, the emergence of this behavior is not guaranteed; it depends on sufficient pretraining task diversity (Raventos et al., 2023). This algorithmic paradigm also extends to linear autoregressive processes, where transformers have been shown to implement a gradient descent step to learn the transition matrix in-context (Sander et al., 2024). In a complementary direction, Lu et al. (2025) develop an asymptotic theory for in-context learning by linear attention, providing exact analytical characterizations in the large-dimensional limit.

**In-Context learning, Markov chains and n-gram models** Our work is closely related to the literature analyzing ICL for sequential probabilistic models like n-grams and Markov chains. Foundational work by Yüksel & Flammarion (2025) established formal generalization bounds for next-token prediction on Markovian data, analyzing its sample complexity. Regarding transformers, several mechanistic studies have investigated how they implement learning algorithms for these models. For bigrams, transformers have been shown to develop induction heads that function like associative memories (Bietti et al., 2023b) and accurately compute posterior probabilities from statistical cues (Edelman et al., 2024). For first order Markov chains, Nichani et al. (2024) demonstrated that transformers learn the causal structure with gradient descent and implement induction heads to estimate the transition probabilities in-context, effectively implementing a Bayes-optimal estimator. This analysis was later extended to higher-order chains (Chen et al., 2024), while the work of D'Angelo et al. (2025) shows that transformers can even learn to select the correct Markov causal structure at inference time. Further theoretical results have explored the transformer loss landscape in this setting (Makkuva et al., 2024), characterized in-context n-grams as near-stationary points (Varre et al., 2025), and shown that constant-depth transformers are sufficient to learn k-th order Markov chains (Rajaraman et al., 2024).

**Transformers and sequential models** Beyond specific learning algorithms, a broader line of work has explored the fundamental capabilities and limitations of transformers as sequential models. On the foundational side, transformers have been shown to be universal approximators of sequence-to-sequence functions (Yun et al., 2020) and even Turing-complete under certain assumptions (Pérez et al., 2021), while recent work has investigated the trade-off between depth and parallel computation (Sanford et al., 2024). In terms of representational power, transformers with sparse attention have been shown to be capable of exactly representing any n-gram model (Svete & Cotterell, 2024).

However, this expressive power has limits; for instance, transformers may be less effective at learning certain Hidden Markov Models (HMMs) compared to RNNs (Hu et al., 2024). Interestingly, when investigating the internal representations that enable inference in HMMs, Shai et al. (2024) showed that transformers maintain interpretable belief states that are linearly encoded in the residual stream. A strength of transformers is their ability to perform in-context model selection. It has been demonstrated that a single transformer can adaptively choose between different base algorithms or even qualitatively different tasks (e.g., regression vs. classification) based on the prompt (Bai et al., 2023a), effectively selecting between different function classes in-context (Yadlowsky et al., 2023). While high-level n-gram statistics can approximate transformer predictions, the mechanism for how the correct "rule" is selected in-context remains an open question (Nguyen, 2024).

## D    ADDITIONAL EXPERIMENTS

In this section we repeat the main-text experiments for different MTD orders, comparing the trained and constructed transformers: in addition to the $m = 4$ case in the main text, we report results for $m = 3$ and $m = 5$. In Figure 6, we plot the KL divergence to the ground-truth transition probabilities across sequence lengths for the trained transformers, the constructed transformer, and the one-step MD estimator (orders 3 and 5). In Figure 7, we instead compare the learned first-layer attention (softmax) to the true transition matrices for orders 3 and 5, with the average row-wise KL divergence reported directly in each panel. Finally, Figure 8 reports attention grids for the trained and constructed transformers (disentangled) at sequence length 64 for MTD orders $m = 3$ and $m = 5$, analogous to the attention maps shown in Figure 4 (left) in the main text.

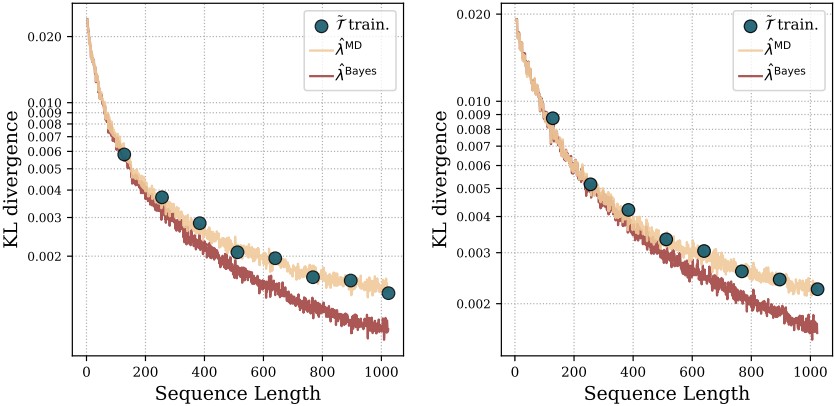

Figure 6: **KL divergence to the ground-truth** We report the KL divergence to the ground truth transition probabilities for the trained transformers, the constructed transformer, and the one-step MD estimator across sequence lengths **Left:** order 3. **Right:** order 5.

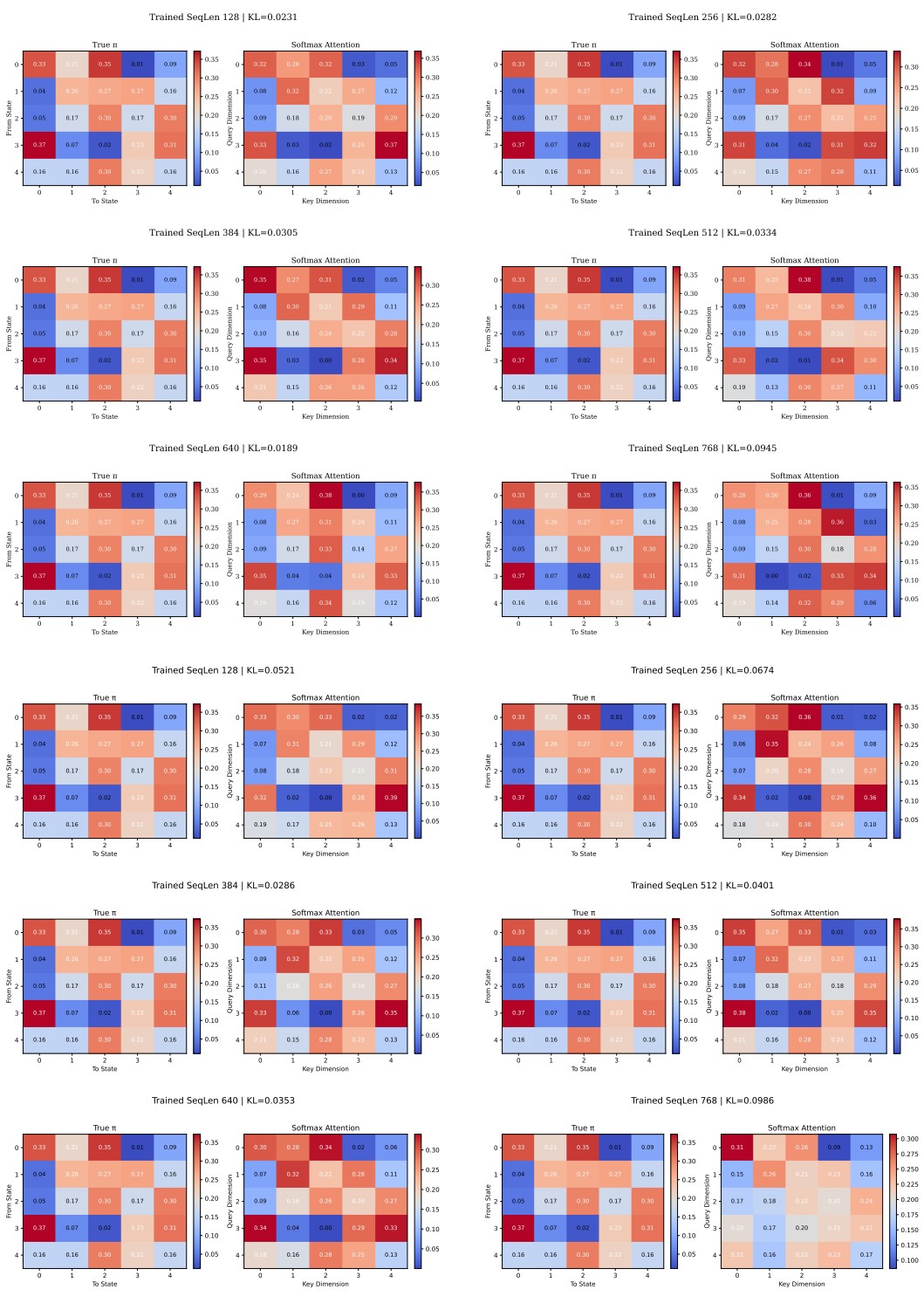

Figure 7: **Layer-1 Attention Softmax vs. True Transition matrices for orders 3 and 5.** For both orders ($m = 3$ top 3 rows, $m = 5$ bottom 3 rows), each panel reports the learned first-layer attention (softmax of $\boldsymbol{W}_1^{A\top}$) alongside the true transition matrix; the average row-wise KL divergence is reported in the panel title. Sequence lengths increase left-to-right, top-to-bottom.

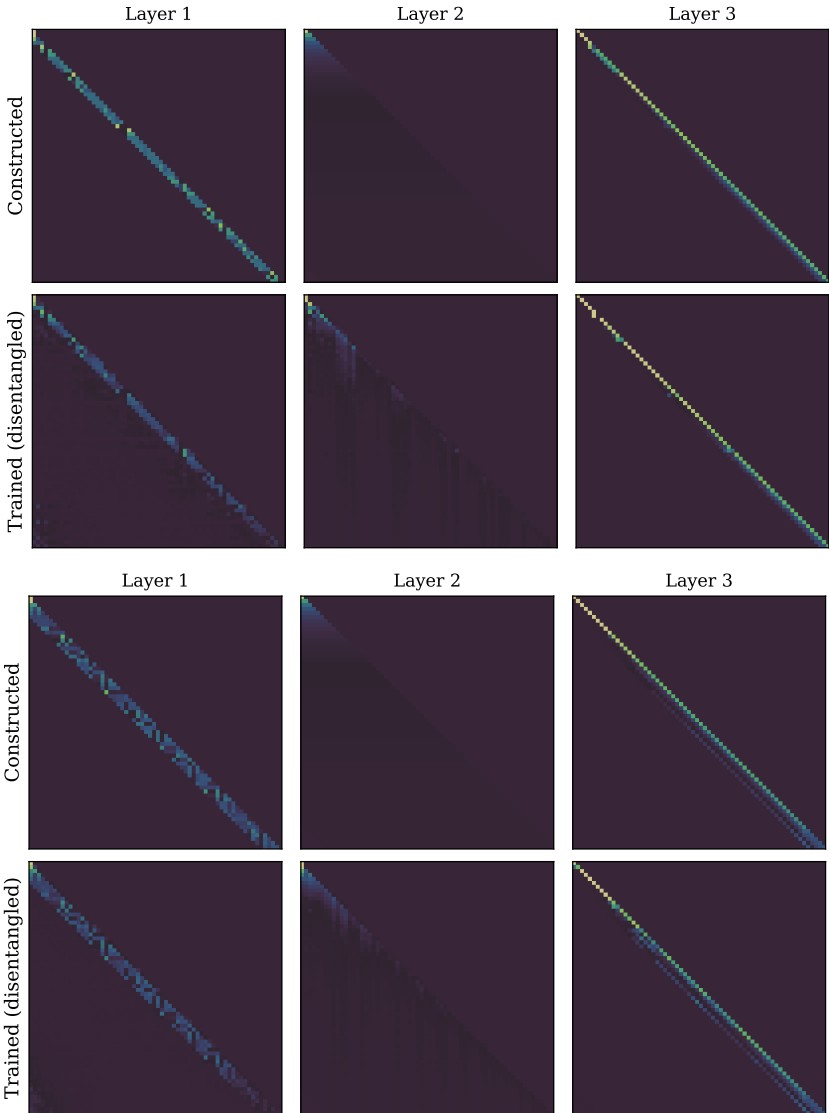

Figure 8: **Comparison of Trained and Constructed Transformers (attention grids). Top:** Attention maps of the trained transformer (disentangled) versus our theoretical construction (seq. length 64, MTD order $m = 3$). **Bottom:** Same as top but for MTD order $m = 5$.

# E   DERIVATION OF THE BAYES-OPTIMAL MTD PREDICTOR

Here, we provide the full derivation for the Bayes-optimal predictive distribution stated in Proposition 1.

Our goal is to derive the predictive distribution for the next state, $p(Y_{t+1}|\boldsymbol{y}_1^t, \boldsymbol{\alpha})$, given an observed data prefix $\boldsymbol{y}_1^t = (y_1, \ldots, y_t)$. The unknown mixture weights $\boldsymbol{\lambda} = (\lambda_1, \ldots, \lambda_m)$ are assumed to be drawn from a Dirichlet prior distribution:

$$p(\boldsymbol{\lambda}|\boldsymbol{\alpha}) = \text{Dirichlet}(\boldsymbol{\lambda}|\boldsymbol{\alpha}) = \frac{\Gamma(\sum_{g=1}^m \alpha_g)}{\prod_{g=1}^m \Gamma(\alpha_g)} \prod_{g=1}^m \lambda_g^{\alpha_g - 1}.$$

The likelihood of the observed data given the parameters $\boldsymbol{\lambda}$ is defined by the MTD model:

$$p(\boldsymbol{y}_1^t \mid \boldsymbol{\lambda}) = \prod_{k=m+1}^t p(y_k \mid \boldsymbol{y}_1^{k-1}, \boldsymbol{\lambda}) = \prod_{k=m+1}^t \left(\sum_{h=1}^m \lambda_h \pi(y_{k-h}, y_k)\right).$$

Combining the likelihood and prior via Bayes' theorem yields the posterior distribution over the mixture weights:

$$p(\boldsymbol{\lambda} \mid \boldsymbol{y}_1^t, \boldsymbol{\alpha}) \propto p(\boldsymbol{y}_1^t \mid \boldsymbol{\lambda}) \cdot p(\boldsymbol{\lambda} \mid \boldsymbol{\alpha}).$$

The Bayesian predictive distribution is formulated by marginalizing the single-step prediction $p(Y_{t+1} = j \mid \boldsymbol{y}_1^t, \boldsymbol{\lambda})$ over this posterior distribution of $\boldsymbol{\lambda}$:

$$p(Y_{t+1} = j \mid \boldsymbol{y}_1^t, \boldsymbol{\alpha}) = \int_{\Delta_{m-1}} p(Y_{t+1} = j \mid \boldsymbol{y}_1^t, \boldsymbol{\lambda}) \cdot p(\boldsymbol{\lambda} \mid \boldsymbol{y}_1^t, \boldsymbol{\alpha}) \, d\boldsymbol{\lambda},$$

where $\Delta_{m-1}$ is the probability simplex. The single-step prediction is simply the MTD model definition:

$$p(Y_{t+1} = j \mid \boldsymbol{y}_1^t, \boldsymbol{\lambda}) = \sum_{g=1}^m \lambda_g \pi(y_{t+1-g}, j).$$

Substituting this into the integral gives:

$$p(Y_{t+1} = j \mid \boldsymbol{y}_1^t, \boldsymbol{\alpha}) = \int_{\Delta_{m-1}} \left(\sum_{g=1}^m \lambda_g \pi(y_{t+1-g}, j)\right) p(\boldsymbol{\lambda} \mid \boldsymbol{y}_1^t, \boldsymbol{\alpha}) \, d\boldsymbol{\lambda}.$$

By the linearity of expectation (and integration), the integral and the finite sum can be interchanged:

$$p(Y_{t+1} = j \mid \boldsymbol{y}_1^t, \boldsymbol{\alpha}) = \sum_{g=1}^m \pi(y_{t+1-g}, j) \left(\int_{\Delta_{m-1}} \lambda_g \cdot p(\boldsymbol{\lambda} \mid \boldsymbol{y}_1^t, \boldsymbol{\alpha}) \, d\boldsymbol{\lambda}\right).$$

We recognize the term in the parentheses as the definition of the posterior mean of the parameter $\lambda_g$:

$$\hat{\lambda}_g^{\text{Bayes}} := \mathbb{E}[\lambda_g \mid \boldsymbol{y}_1^t, \boldsymbol{\alpha}] = \int_{\Delta_{m-1}} \lambda_g \cdot p(\boldsymbol{\lambda} \mid \boldsymbol{y}_1^t, \boldsymbol{\alpha}) \, d\boldsymbol{\lambda}.$$

This substitution yields the final form of the Bayes-optimal predictor, completing the proof:

$$p(Y_{t+1} = j \mid \boldsymbol{y}_1^t, \boldsymbol{\alpha}) = \sum_{g=1}^m \hat{\lambda}_g^{\text{Bayes}} \cdot \pi(y_{t+1-g}, j).$$

## E.1   THE STRUCTURE OF THE BAYES-OPTIMAL ESTIMATOR

While the posterior mean is intractable to compute, its structure can be derived exactly. In particular it can be shown that the MTD posterior is a finite mixture of Dirichlet distributions, and from this, we can derive that the mean of the posterior preserves the classic 'add-constant' structure of conjugate Bayesian models, where the unobserved data counts are replaced by their posterior expectation.

**Proposition 4** (Posterior as a Mixture of Dirichlets)*. Given the MTD likelihood $L(\boldsymbol{\lambda}) = p(\boldsymbol{y}_1^t \mid \boldsymbol{\lambda})$ and a Dirichlet prior $p(\boldsymbol{\lambda} \mid \boldsymbol{\alpha}) = Dir(\boldsymbol{\lambda} \mid \boldsymbol{\alpha})$, the posterior distribution is a finite mixture of Dirichlet distributions:*

$$p(\boldsymbol{\lambda} \mid \boldsymbol{y}_1^t, \boldsymbol{\alpha}) = \sum_{z \in \{1, \ldots, m\}^{t-m}} \pi(z) \cdot Dir(\boldsymbol{\lambda} \mid \boldsymbol{\alpha} + k(z)), \tag{9}$$

*where $z = (z_{m+1}, \ldots, z_t)$ is a latent assignment path, $k(z)$ is the vector of counts of each lag in path $\boldsymbol{z}$, and $\pi(z) = p(Z = z \mid \boldsymbol{y}_1^t, \boldsymbol{\alpha})$ are the true posterior probabilities of the latent paths.*

*Proof.* We begin with Bayes' theorem for the posterior distribution:

$$p(\boldsymbol{\lambda} \mid \boldsymbol{y}_1^t, \boldsymbol{\alpha}) \propto p(\boldsymbol{y}_1^t \mid \boldsymbol{\lambda}) \cdot p(\boldsymbol{\lambda} \mid \boldsymbol{\alpha}).$$

The central idea is to express the observed-data likelihood, $p(\boldsymbol{y}_1^t \mid \boldsymbol{\lambda})$, by marginalizing over all possible latent assignment paths $\boldsymbol{z}$. A path $\boldsymbol{z} = (z_{m+1}, \ldots, z_t)$ specifies which lag was used at each step $k$.

$$p(\boldsymbol{y}_1^t \mid \boldsymbol{\lambda}) = \sum_{\boldsymbol{z} \in \{1, \ldots, m\}^{t-m}} p(\boldsymbol{y}_1^t, \boldsymbol{z} \mid \boldsymbol{\lambda}).$$

The joint probability of the data and a specific path $\boldsymbol{z}$, known as the complete-data likelihood, is given by:

$$\begin{aligned}
p(\boldsymbol{y}_1^t, z \mid \boldsymbol{\lambda}) &= \prod_{k=m+1}^{t} p(y_k, z_k \mid \boldsymbol{y}_1^{k-1}, \boldsymbol{\lambda}) \\
&= \prod_{k=m+1}^{t} p(z_k \mid \boldsymbol{\lambda}) \cdot p(y_k \mid \boldsymbol{y}_1^{k-1}, z_k, \boldsymbol{\lambda}) \\
&= \prod_{k=m+1}^{t} \lambda_{z_k} \cdot \pi(y_{k-z_k}, y_k).
\end{aligned}$$

We can group the terms that depend on $\boldsymbol{\lambda}$ and those that do not. Let $k_g(z) = \sum_{k=m+1}^{t} \mathbb{I}(z_k = g)$ be the number of times lag $g$ is used in path $z$. Then:

$$p(\boldsymbol{y}_1^t, z \mid \boldsymbol{\lambda}) = \left( \prod_{g=1}^{m} \lambda_g^{k_g(z)} \right) \left( \prod_{k=m+1}^{t} \pi(y_{k-z_k}, y_k) \right).$$

Let $P(\boldsymbol{y} \mid z) := \prod_{k=m+1}^{t} \pi(y_{k-z_k}, y_k)$, which is constant with respect to $\boldsymbol{\lambda}$. The prior is given by $p(\boldsymbol{\lambda} \mid \boldsymbol{\alpha}) = \frac{1}{B(\boldsymbol{\alpha})} \prod_{g=1}^{m} \lambda_g^{\alpha_g - 1}$, where $B(\boldsymbol{\alpha})$ is the multivariate beta function. Substituting these into the expression for the posterior:

$$\begin{aligned}
p(\boldsymbol{\lambda} \mid \boldsymbol{y}_1^t, \boldsymbol{\alpha}) &\propto \left( \sum_z P(\boldsymbol{y} \mid z) \prod_{g=1}^{m} \lambda_g^{k_g(z)} \right) \left( \frac{1}{B(\boldsymbol{\alpha})} \prod_{g=1}^{m} \lambda_g^{\alpha_g - 1} \right) \\
&\propto \sum_z P(\boldsymbol{y} \mid z) \prod_{g=1}^{m} \lambda_g^{\alpha_g + k_g(z) - 1}.
\end{aligned}$$

We recognize that the term $\prod_g \lambda_g^{(\alpha_g + k_g(z)) - 1}$ is the kernel of a Dirichlet distribution, $\mathrm{Dir}(\boldsymbol{\lambda} \mid \boldsymbol{\alpha} + k(z))$. We can write it as $B(\boldsymbol{\alpha} + k(z)) \cdot \mathrm{Dir}(\boldsymbol{\lambda} \mid \boldsymbol{\alpha} + k(z))$. Thus, the posterior takes the form of a weighted sum of Dirichlet densities:

$$p(\boldsymbol{\lambda} \mid \boldsymbol{y}_1^t, \boldsymbol{\alpha}) = \sum_z \pi(z) \cdot \mathrm{Dir}(\boldsymbol{\lambda} \mid \boldsymbol{\alpha} + k(z)),$$

where the mixture weights $\pi(z)$ are the normalized coefficients, which are precisely the true posterior probabilities of the latent paths, $p(Z = z \mid \boldsymbol{y}_1^t, \boldsymbol{\alpha})$. This completes the proof. $\square$

From this mixture structure, we can derive an exact identity for the posterior mean.

**Proposition 5** (Bayes Mean as Add-Constant-to-Expected-Counts). *The components of the Bayesian posterior mean, $\hat{\boldsymbol{\lambda}}^{Bayes} = \mathbb{E}[\boldsymbol{\lambda} \mid \boldsymbol{y}_1^t, \boldsymbol{\alpha}]$, are given by:*

$$\hat{\lambda}_g^{Bayes} = \frac{\alpha_g + \mathbb{E}_{Z \sim \pi}[k_g(Z)]}{\alpha_0 + (t - m)}, \tag{10}$$

*where $\alpha_0 = \sum_j \alpha_j$, and $\mathbb{E}_{Z \sim \pi}[k_g(Z)]$ is the posterior expected number of times lag $g$ was used, which is computationally intractable.*

*Proof.* The posterior mean is defined by the integral of $\boldsymbol{\lambda}$ over its posterior distribution:

$$\hat{\boldsymbol{\lambda}}^{Bayes} = \int_{\Delta_{m-1}} \boldsymbol{\lambda} \cdot p(\boldsymbol{\lambda} \mid \boldsymbol{y}_1^t, \boldsymbol{\alpha}) \, d\boldsymbol{\lambda}.$$

We substitute the mixture-of-Dirichlets form of the posterior from Proposition 4:

$$\hat{\boldsymbol{\lambda}}^{Bayes} = \int_{\Delta_{m-1}} \boldsymbol{\lambda} \cdot \left( \sum_z \pi(z) \cdot \mathrm{Dir}(\boldsymbol{\lambda} \mid \boldsymbol{\alpha} + k(z)) \right) d\boldsymbol{\lambda}.$$

Since the summation is over a finite set of paths $z$, we can swap the integral and the summation:

$$\hat{\boldsymbol{\lambda}}^{Bayes} = \sum_z \pi(z) \left( \int_{\Delta_{m-1}} \boldsymbol{\lambda} \cdot \mathrm{Dir}(\boldsymbol{\lambda} \mid \boldsymbol{\alpha} + k(z)) \, d\boldsymbol{\lambda} \right).$$

The term inside the parentheses is the definition of the mean of a Dirichlet distribution with parameter vector $\boldsymbol{\beta} = \boldsymbol{\alpha} + k(z)$. The mean of a $\mathrm{Dir}(\boldsymbol{\beta})$ distribution is the vector $\boldsymbol{\beta}/\beta_0$, where $\beta_0 = \sum_j \beta_j$. In our case, the sum of the parameters is:

$$\sum_{g=1}^m (\alpha_g + k_g(z)) = \left( \sum_g \alpha_g \right) + \left( \sum_g k_g(z) \right) = \alpha_0 + (t - m).$$

Therefore, the inner integral evaluates to the vector $\frac{\boldsymbol{\alpha} + k(z)}{\alpha_0 + (t-m)}$. Substituting this back:

$$\hat{\boldsymbol{\lambda}}^{Bayes} = \sum_z \pi(z) \frac{\boldsymbol{\alpha} + k(z)}{\alpha_0 + (t - m)}.$$

This expression is an expectation over the posterior distribution of latent paths, $Z \sim \pi(z)$. We can write it as:

$$\hat{\boldsymbol{\lambda}}^{Bayes} = \mathbb{E}_{Z \sim \pi} \left[ \frac{\boldsymbol{\alpha} + k(Z)}{\alpha_0 + (t - m)} \right].$$

By the linearity of expectation, we can take the expectation inside for each component $g$:

$$\hat{\lambda}_g^{Bayes} = \frac{\mathbb{E}_{Z \sim \pi}[\alpha_g + k_g(Z)]}{\alpha_0 + (t - m)} = \frac{\alpha_g + \mathbb{E}_{Z \sim \pi}[k_g(Z)]}{\alpha_0 + (t - m)}.$$

This reveals the "add-constant-to-expected-counts" structure of the estimator, completing the proof. $\qquad \square$

## F    APPROXIMATIONS FOR THE MEAN OF THE POSTERIOR DISTRIBUTION

In this section, we outline the methods used to approximate the mean of the posterior distribution over the mixture weights $\boldsymbol{\lambda}$.

### F.1    MARKOV CHAIN MONTE CARLO (MCMC)

Since an analytical solution is unavailable, we approximate the Bayesian predictive distribution using samples from the posterior distribution generated by a Markov Chain Monte Carlo (MCMC) method, specifically **Gibbs sampling**. Gibbs sampling is well-suited for this problem because, while the full posterior is intractable, the conditional posteriors of the parameters and latent variables are simple to sample from.

The procedure involves augmenting the model with latent variables $\boldsymbol{Z}_1^t = (Z_{m+1}, \ldots, Z_t)$, where $Z_k = g$ indicates that lag $g$ was used to generate the transition to $Y_k$. The Gibbs sampler iteratively draws from the two full conditional distributions:

1. **Sample Latent Variables $Z$ given Parameters $\boldsymbol{\lambda}$:** For a given parameter vector $\boldsymbol{\lambda}^{(k-1)}$ from the previous iteration, we sample each latent variable $Z_s$ for $s \in \{m+1, \ldots, t\}$ from its categorical conditional posterior:

$$\mathbb{P}(Z_s = g \mid \boldsymbol{y}_1^t, \boldsymbol{\lambda}^{(k-1)}) = \frac{\lambda_g^{(k-1)} \pi(y_{s-g}, y_s)}{\sum_{h=1}^m \lambda_h^{(k-1)} \pi(y_{s-h}, y_s)}. \tag{11}$$

   This provides a complete sampled sequence of lags, $\boldsymbol{z}^{(k)} = (z_{m+1}^{(k)}, \ldots, z_t^{(k)})$.

2. **Sample Parameters $\boldsymbol{\lambda}$ given Latent Variables $Z$:** Given the sampled lags $\boldsymbol{z}^{(k)}$, the Dirichlet prior is now conjugate to the complete-data likelihood. We first count the occurrences of each lag, $n_g = \sum_{s=m+1}^t \mathbb{I}(z_s^{(k)} = g)$. The full conditional posterior for $\boldsymbol{\lambda}$ is a Dirichlet distribution, from which we draw the next sample $\boldsymbol{\lambda}^{(k)}$:

$$p(\boldsymbol{\lambda} \mid \boldsymbol{y}_1^t, \boldsymbol{z}^{(k)}, \boldsymbol{\alpha}) = \text{Dirichlet}(\boldsymbol{\lambda} \mid \alpha_1 + n_1, \ldots, \alpha_m + n_m). \tag{12}$$

After running the sampler for $K_{\text{total}}$ iterations and discarding an initial burn-in period, we obtain a set of $K$ samples, $\{\boldsymbol{\lambda}^{(1)}, \boldsymbol{\lambda}^{(2)}, \ldots, \boldsymbol{\lambda}^{(K)}\}$, that are approximately drawn from the true posterior $p(\boldsymbol{\lambda} \mid \boldsymbol{y}_1^t, \boldsymbol{\alpha})$ shown in Equation 4.

The integral in Equation 4 is then approximated via a Monte Carlo average:

$$\hat{p}(Y_{t+1} = j \mid \boldsymbol{y}_1^t) = \frac{1}{K} \sum_{k=1}^K p(Y_{t+1} = j \mid \boldsymbol{y}_1^t, \boldsymbol{\lambda}^{(k)})$$

$$= \frac{1}{K} \sum_{k=1}^K \left( \sum_{g=1}^m \lambda_g^{(k)} \pi(y_{t+1-g}, j) \right). \tag{13}$$

This estimate converges to the true Bayes optimal predictive distribution as $K \to \infty$. It represents the theoretical performance limit for inference under the MTD model assumptions, providing a gold-standard benchmark against which other estimators can be compared.

## G   ALGORITHMS FOR MAXIMUM LIKELIHOOD ESTIMATION OF MTD

Maximum Likelihood Estimation (MLE) for the Mixture Transition Distribution (MTD) does not admit a closed-form solution, therefore iterative optimization algorithms are required. This section details two iterative algorithms suited for this task. We first review the Expectation-Maximization (EM) algorithm, a standard and widely-used method for latent variable models. We then present Mirror Descent (MD), an alternative optimization framework that is central to our work.

### G.1   EXPECTATION-MAXIMIZATION (EM) ALGORITHM

The Expectation-Maximization (EM) algorithm is a widely-used iterative method for finding Maximum Likelihood estimates in statistical models with latent variables. In the context of the MTD model, the latent variables correspond to the specific mixture component responsible for generating each observation. The algorithm alternates between two steps: the Expectation (E) step, where it computes the expected log-likelihood with respect to the posterior distribution of the latent variables, and the Maximization (M) step, where it updates the model parameters to maximize this expected value.

Given the latent variables $\boldsymbol{Z} = (Z_{m+1}, \ldots, Z_T)$, where each $Z_t \in \{1, \ldots, m\}$. The variable $Z_t = g$ indicates that the $g^{th}$ mixture component (corresponding to lag $g$) was responsible for generating the transition to $Y_t$ at time $t$. The complete data are $(\boldsymbol{y}, \boldsymbol{z})$.

The likelihood of the complete data $(\boldsymbol{y}_{m+1}^T, \boldsymbol{z}_{m+1}^T)$, conditional on $\boldsymbol{y}_1^m$, is given by:

$$\mathbb{P}(\boldsymbol{y}_{m+1}^T, \boldsymbol{z}_{m+1}^T \mid \boldsymbol{y}_1^m; \boldsymbol{\lambda}) = \prod_{t=m+1}^T \mathbb{P}(y_t, z_t \mid \boldsymbol{y}_1^{t-1}; \boldsymbol{\lambda})$$

$$= \prod_{t=m+1}^T \mathbb{P}(Z_t = z_t \mid \boldsymbol{y}_1^{t-1}; \boldsymbol{\lambda}) \mathbb{P}(y_t \mid Z_t = z_t, \boldsymbol{y}_1^{t-1}; \boldsymbol{\lambda}).$$

Under the MTD model assumptions:

- $\mathbb{P}(Z_t = g \mid \boldsymbol{y}_1^{t-1}; \boldsymbol{\lambda}) = \lambda_g$
- $\mathbb{P}(y_t \mid Z_t = g, \boldsymbol{y}_1^{t-1}; \boldsymbol{\lambda}) = \pi(y_{t-g}, y_t)$

Thus, $\mathbb{P}(y_t, Z_t = g \mid \boldsymbol{y}_1^{t-1}; \boldsymbol{\lambda}) = \lambda_g \pi(y_{t-g}, y_t)$. The complete data likelihood becomes:

$$\mathbb{P}(\boldsymbol{y}_{m+1}^T, \boldsymbol{z}_{m+1}^T \mid \boldsymbol{y}_1^m; \boldsymbol{\lambda}) = \prod_{t=m+1}^T \lambda_{z_t} \pi(y_{t-z_t}, y_t).$$

The complete data log-likelihood, $\ell_c(\boldsymbol{\lambda}; \boldsymbol{y}, \boldsymbol{z}) = \log \mathbb{P}(\boldsymbol{y}_{m+1}^T, \boldsymbol{z}_{m+1}^T \mid \boldsymbol{y}_1^m; \boldsymbol{\lambda})$, is:

$$\ell_c(\boldsymbol{\lambda}; \boldsymbol{y}, \boldsymbol{z}) = \sum_{t=m+1}^T \log(\lambda_{z_t} \pi_{z_t}(y_{t-z_t}, y_t))$$

$$= \sum_{t=m+1}^T \sum_{g=1}^m \mathbb{I}(z_t = g) \log(\lambda_g \pi(y_{t-g}, y_t)), \tag{14}$$

where $\mathbb{I}(\cdot)$ is the indicator function.

## G.2 EM ALGORITHM STEPS

Let $\boldsymbol{\lambda}^{(k)}$ be the estimate of $\boldsymbol{\lambda}$ at iteration $k$.

**E-Step (Expectation)** The E-step computes the expectation of the complete-data log-likelihood equation 14 with respect to the conditional distribution of the latent variables $\boldsymbol{Z}$ given the observed data $\boldsymbol{y}$ and the current parameter estimate $\boldsymbol{\lambda}^{(k)}$. This expectation defines the $Q$ function:

$$Q(\boldsymbol{\lambda} \mid \boldsymbol{\lambda}^{(k)}) = \mathbb{E}_{\boldsymbol{Z} \mid \boldsymbol{y}, \boldsymbol{\lambda}^{(k)}}[\ell_c(\boldsymbol{\lambda}; \boldsymbol{y}, \boldsymbol{Z})]$$

$$= \mathbb{E}\left[ \sum_{t=m+1}^T \sum_{g=1}^m \mathbb{I}(Z_t = g) \log(\lambda_g \pi(y_{t-g}, y_t)) \,\middle|\, \boldsymbol{y}, \boldsymbol{\lambda}^{(k)} \right]$$

$$= \sum_{t=m+1}^T \sum_{g=1}^m \mathbb{E}[\mathbb{I}(Z_t = g) \mid \boldsymbol{y}, \boldsymbol{\lambda}^{(k)}] \log(\lambda_g \pi(y_{t-g}, y_t)).$$

The core computation is the posterior probability (responsibility) of $Z_t = g$:

$$\gamma_t^{(k)}(g) := \mathbb{E}[\mathbb{I}(Z_t = g) \mid \boldsymbol{y}, \boldsymbol{\lambda}^{(k)}] = \mathbb{P}(Z_t = g \mid \boldsymbol{y}, \boldsymbol{\lambda}^{(k)}).$$

Due to the MTD model structure, future observations $\boldsymbol{y}_{t+1}^T$ are conditionally independent of $Z_t$ given $\boldsymbol{y}_1^t$. Thus, the posterior probability simplifies:

$$\mathbb{P}(Z_t = g \mid \boldsymbol{y}, \boldsymbol{\lambda}^{(k)}) = \mathbb{P}(Z_t = g \mid \boldsymbol{y}_1^t, \boldsymbol{\lambda}^{(k)}).$$

Using Bayes' theorem:

$$\gamma_t^{(k)}(g) = \mathbb{P}(Z_t = g \mid \boldsymbol{y}_1^t, \boldsymbol{\lambda}^{(k)})$$

$$= \frac{\mathbb{P}(y_t \mid Z_t = g, \boldsymbol{y}_1^{t-1}, \boldsymbol{\lambda}^{(k)}) \mathbb{P}(Z_t = g \mid \boldsymbol{y}_1^{t-1}, \boldsymbol{\lambda}^{(k)})}{\mathbb{P}(y_t \mid \boldsymbol{y}_1^{t-1}, \boldsymbol{\lambda}^{(k)})}$$

$$= \frac{\pi(y_{t-g}, y_t) \lambda_g^{(k)}}{\sum_{h=1}^m \mathbb{P}(y_t, Z_t = h \mid \boldsymbol{y}_1^{t-1}, \boldsymbol{\lambda}^{(k)})}$$

$$= \frac{\lambda_g^{(k)} \pi(y_{t-g}, y_t)}{\sum_{h=1}^m \lambda_h^{(k)} \pi(y_{t-h}, y_t)}. \tag{15}$$

The E-step involves calculating these responsibilities $\gamma_t^{(k)}(g)$ for all $t \in \{m+1, \ldots, T\}$ and $g \in \{1, \ldots, m\}$. The $Q$ function is then:

$$Q(\boldsymbol{\lambda} \mid \boldsymbol{\lambda}^{(k)}) = \sum_{t=m+1}^T \sum_{g=1}^m \gamma_t^{(k)}(g)(\log \lambda_g + \log \pi(y_{t-g}, y_t)). \tag{16}$$

**M-Step (Maximization)** The M-step finds the parameter values $\boldsymbol{\lambda}$ that maximize the $Q$ function equation 16 subject to the constraints $\lambda_g \geq 0$ and $\sum_{g=1}^{m} \lambda_g = 1$. This gives the updated estimate $\boldsymbol{\lambda}^{(k+1)}$.

$$\boldsymbol{\lambda}^{(k+1)} = \arg\max_{\boldsymbol{\lambda}} Q(\boldsymbol{\lambda} \mid \boldsymbol{\lambda}^{(k)}).$$

Since the terms $\log \pi(y_{t-g}, y_t)$ do not depend on $\boldsymbol{\lambda}$, we maximize:

$$f(\boldsymbol{\lambda}) = \sum_{t=m+1}^{T} \sum_{g=1}^{m} \gamma_t^{(k)}(g) \log \lambda_g = \sum_{g=1}^{m} \left( \sum_{t=m+1}^{T} \gamma_t^{(k)}(g) \right) \log \lambda_g.$$

Let $C_g = \sum_{t=m+1}^{T} \gamma_t^{(k)}(g)$. We maximize $f(\boldsymbol{\lambda}) = \sum_{g=1}^{m} C_g \log \lambda_g$ subject to $\sum_{g=1}^{m} \lambda_g = 1$. We use a Lagrange multiplier $\mu$:

$$\mathcal{L}(\boldsymbol{\lambda}, \mu) = \sum_{g=1}^{m} C_g \log \lambda_g - \mu \left( \sum_{g=1}^{m} \lambda_g - 1 \right).$$

Setting partial derivatives to zero:

$$\frac{\partial \mathcal{L}}{\partial \lambda_g} = \frac{C_g}{\lambda_g} - \mu = 0 \implies \lambda_g = \frac{C_g}{\mu}$$

$$\frac{\partial \mathcal{L}}{\partial \mu} = - \left( \sum_{g=1}^{m} \lambda_g - 1 \right) = 0 \implies \sum_{g=1}^{m} \lambda_g = 1.$$

Substituting $\lambda_g = C_g/\mu$ into the constraint yields $\mu = \sum_{h=1}^{m} C_h$. Therefore:

$$\lambda_g = \frac{C_g}{\sum_{h=1}^{m} C_h} = \frac{\sum_{t=m+1}^{T} \gamma_t^{(k)}(g)}{\sum_{h=1}^{m} \sum_{t'=m+1}^{T} \gamma_{t'}^{(k)}(h)}.$$

The denominator simplifies as $\sum_{h=1}^{m} \sum_{t'=m+1}^{T} \gamma_{t'}^{(k)}(h) = \sum_{t'=m+1}^{T} \sum_{h=1}^{m} \gamma_{t'}^{(k)}(h) = \sum_{t'=m+1}^{T} 1 = T - m$. The M-step update rule is thus:

$$\lambda_g^{(k+1)} = \frac{\sum_{t=m+1}^{T} \gamma_t^{(k)}(g)}{T - m}. \tag{17}$$

### G.3 SUMMARY OF THE EM ALGORITHM

The EM algorithm for estimating the mixture weights $\boldsymbol{\lambda}$ in the MTD model, assuming known transition matrices $\boldsymbol{\pi}_g$, proceeds as follows:

1. **Initialization:** Choose initial weights $\boldsymbol{\lambda}^{(0)} = (\lambda_1^{(0)}, \ldots, \lambda_m^{(0)})$ such that $\lambda_g^{(0)} \geq 0$ for all $g \in \{1, \ldots, m\}$ and $\sum_{g=1}^{m} \lambda_g^{(0)} = 1$. Set the iteration counter $k = 0$.

2. **E-Step (Expectation):** Compute the responsibilities $\gamma_t^{(k)}(g)$ for each time point $t \in \{m+1, \ldots, T\}$ and each mixture component $g \in \{1, \ldots, m\}$, using the current parameter estimates $\boldsymbol{\lambda}^{(k)}$:

$$\gamma_t^{(k)}(g) = \frac{\lambda_g^{(k)} \pi(y_{t-g}, y_t)}{\sum_{h=1}^{m} \lambda_h^{(k)} \pi(y_{t-h}, y_t)}. \tag{18}$$

3. **M-Step (Maximization):** Update the mixture weights $\lambda_g^{(k+1)}$ for each component $g \in \{1, \ldots, m\}$ using the computed responsibilities:

$$\lambda_g^{(k+1)} = \frac{\sum_{t=m+1}^{T} \gamma_t^{(k)}(g)}{T - m}. \tag{19}$$

4. **Convergence Check:** If the change in the parameter estimates (e.g., $\|\boldsymbol{\lambda}^{(k+1)} - \boldsymbol{\lambda}^{(k)}\|$) or the change in the observed data log-likelihood (e.g., $\ell(\boldsymbol{\lambda}^{(k+1)}; \boldsymbol{y}) - \ell(\boldsymbol{\lambda}^{(k)}; \boldsymbol{y})$, where $\ell(\boldsymbol{\lambda}; \boldsymbol{y})$ is the observed data log-likelihood) is below a predefined tolerance $\epsilon$, stop the algorithm and return $\hat{\boldsymbol{\lambda}} = \boldsymbol{\lambda}^{(k+1)}$ as the estimated mixture weights. Otherwise, set $k \leftarrow k + 1$ and repeat from Step 2.

### G.4 MIRROR DESCENT

This appendix provides detailed derivations for Equation (5) and Proposition 2.

### G.5 DERIVATION OF THE EXPONENTIATED GRADIENT ALGORITHM

The Exponentiated Gradient (EG) algorithm is a specific instance of the Mirror Descent (MD) online optimization algorithm. We apply it to the problem of maximizing the MTD log-likelihood function, $\ell(\boldsymbol{\lambda})$, with respect to the mixture weights $\boldsymbol{\lambda} = (\lambda_1, \ldots, \lambda_m)$. The weights are constrained to the probability simplex, $\Delta_{m-1} = \{\boldsymbol{\lambda} \in \mathbb{R}^m \mid \sum_g \lambda_g = 1, \lambda_g \geq 0\}$.

The general MD update step at iteration $k$ linearizes the objective function around the current estimate $\boldsymbol{\lambda}^{(k)}$ and adds a regularization term. The next iterate, $\boldsymbol{\lambda}^{(k+1)}$, is found by solving:

$$\boldsymbol{\lambda}^{(k+1)} = \arg\max_{\boldsymbol{\lambda} \in \Delta_{m-1}} \left\{ \langle \nabla\ell(\boldsymbol{\lambda}^{(k)}), \boldsymbol{\lambda} \rangle - \frac{1}{\eta} D_\Psi(\boldsymbol{\lambda}, \boldsymbol{\lambda}^{(k)}) \right\}, \tag{20}$$

where $\eta > 0$ is the learning rate, $\nabla\ell(\boldsymbol{\lambda}^{(k)})$ is the gradient of the log-likelihood evaluated at $\boldsymbol{\lambda}^{(k)}$, and $D_\Psi$ is the Bregman divergence associated with a potential function $\Psi$. For optimization over the simplex, the standard choice is the negative entropy potential, $\Psi(\boldsymbol{\lambda}) = \sum_{g=1}^m \lambda_g \log \lambda_g$. The resulting Bregman divergence is the unnormalized Kullback-Leibler (KL) divergence:

$$D_\Psi(\boldsymbol{\lambda}, \boldsymbol{\lambda}^{(k)}) = \sum_{g=1}^m \lambda_g \log \frac{\lambda_g}{\lambda_g^{(k)}}. \tag{21}$$

Therefore the optimization problem in Equation (20) becomes:

$$\boldsymbol{\lambda}^{(k+1)} = \arg\max_{\boldsymbol{\lambda} \in \Delta_{m-1}} \left\{ \langle \nabla\ell(\boldsymbol{\lambda}^{(k)}), \boldsymbol{\lambda} \rangle - \frac{1}{\eta} \sum_{g=1}^m \lambda_g \log \frac{\lambda_g}{\lambda_g^{(k)}} \right\}. \tag{22}$$

To solve the optimization problem in Eq. 20, we form the Lagrangian with a multiplier $\mu$ for the constraint $\sum_g \lambda_g = 1$:

$$\mathcal{L}(\boldsymbol{\lambda}, \mu) = \eta\langle \nabla\ell(\boldsymbol{\lambda}^{(k)}), \boldsymbol{\lambda} \rangle - \sum_g \lambda_g \log \frac{\lambda_g}{\lambda_g^{(k)}} - \mu \left( \sum_g \lambda_g - 1 \right). \tag{23}$$

Setting the derivative $\partial\mathcal{L}/\partial\lambda_g$ to zero yields:

$$\eta\nabla_\lambda\ell(\boldsymbol{\lambda}^{(k)})_g - \left( \log \frac{\lambda_g}{\lambda_g^{(k)}} + 1 \right) - \mu = 0$$

$$\log \frac{\lambda_g}{\lambda_g^{(k)}} = \eta\nabla_\lambda\ell(\boldsymbol{\lambda}^{(k)})_g - \mu - 1$$

$$\lambda_g = \lambda_g^{(k)} \exp(\eta\nabla_\lambda\ell(\boldsymbol{\lambda}^{(k)})_g) \exp(-\mu - 1).$$

The term $\exp(-\mu - 1)$ serves as a normalization constant to ensure $\sum_g \lambda_g = 1$. This leads directly to the EG update rule presented in Equation (5):

$$\lambda_g^{(k+1)} = \frac{\lambda_g^{(k)} \exp(\eta \cdot \nabla_\lambda\ell(\boldsymbol{\lambda}^{(k)})_g)}{\sum_{h=1}^m \lambda_h^{(k)} \exp(\eta \cdot \nabla_\lambda\ell(\boldsymbol{\lambda}^{(k)})_h)}. \tag{5, repeated}$$

### G.6 DERIVATION OF THE ONE-STEP MTD ESTIMATOR

We now prove Proposition 2 by specializing the EG update to the MTD model and evaluating it at the uniform prior $\boldsymbol{\lambda}^{(0)} = (1/m, \ldots, 1/m)$.

**Step 1: MTD Log-Likelihood and its Gradient.** Given an observed sequence prefix $\boldsymbol{y}_1^t$, the MTD log-likelihood is:

$$\ell(\boldsymbol{\lambda}) = \log p(\boldsymbol{y}_1^t \mid \boldsymbol{\lambda}) = \sum_{k=m+1}^{t} \log \left( \sum_{h=1}^{m} \lambda_h \pi(y_{k-h}, y_k) \right). \tag{24}$$

The $g$-th component of its gradient is:

$$\nabla_\lambda \ell(\boldsymbol{\lambda})_g = \frac{\partial \ell(\boldsymbol{\lambda})}{\partial \lambda_g} = \sum_{k=m+1}^{t} \frac{\pi(y_{k-g}, y_k)}{\sum_{h=1}^{m} \lambda_h \pi(y_{k-h}, y_k)}. \tag{25}$$

**Step 2: Evaluate Gradient at the Uniform Prior.** We evaluate this gradient at $\boldsymbol{\lambda}^{(0)} = (1/m, \ldots, 1/m)$:

$$\nabla_\lambda \ell(\boldsymbol{\lambda}^{(0)})_g = \sum_{k=m+1}^{t} \frac{\pi(y_{k-g}, y_k)}{\sum_{h=1}^{m}(1/m)\pi(y_{k-h}, y_k)}$$

$$= m \sum_{k=m+1}^{t} \frac{\pi(y_{k-g}, y_k)}{\sum_{h=1}^{m} \pi(y_{k-h}, y_k)}. \tag{26}$$

We recognize the term inside the summation as the posterior responsibility of lag $g$ under the uniform model:

$$\gamma_k(g) := p(Z_k = g \mid \boldsymbol{y}_1^k, \boldsymbol{\lambda}^{(0)}) \tag{27}$$

$$= \frac{p(y_k \mid y_{k-g})p(Z_k = g|\boldsymbol{\lambda}^{(0)})}{\sum_h p(y_k \mid y_{k-h})p(Z_k = h|\boldsymbol{\lambda}^{(0)})} \tag{28}$$

$$= \frac{\pi(y_{k-g}, y_k) \cdot (1/m)}{\sum_h \pi(y_{k-h}, y_k) \cdot (1/m)} \tag{29}$$

$$= \frac{\pi(y_{k-g}, y_k)}{\sum_{h=1}^{m} \pi(y_{k-h}, y_k)}. \tag{30}$$

Thus, the gradient at the uniform prior is a scaled sum of these responsibilities:

$$\nabla_\lambda \ell(\boldsymbol{\lambda}^{(0)})_g = m \sum_{k=m+1}^{t} \gamma_k^{\text{unif}}(g). \tag{31}$$

**Step 3: Apply the EG Update Rule.** Finally, we substitute $\lambda_g^{(0)} = 1/m$ and the derived gradient into the EG update rule (Eq. 5) to find $\lambda_g^{(1)}$:

$$\hat{\boldsymbol{\lambda}}_g^{\text{MD}} = \lambda_g^{(1)} = \frac{\lambda_g^{(0)} \exp\left(\eta \cdot \nabla_\lambda \ell(\boldsymbol{\lambda}^{(0)})_g\right)}{\sum_{j=1}^{m} \lambda_j^{(0)} \exp\left(\eta \cdot \nabla_\lambda \ell(\boldsymbol{\lambda}^{(0)})_j\right)}$$

$$= \frac{(1/m) \cdot \exp\left(\eta \cdot m \sum_{k=m+1}^{t} \gamma_k^{\text{unif}}(g)\right)}{\sum_{j=1}^{m}(1/m) \cdot \exp\left(\eta \cdot m \sum_{k=m+1}^{t} \gamma_k^{\text{unif}}(j)\right)}$$

$$= \frac{\exp\left(\eta m \sum_{k=m+1}^{t} \gamma_k^{\text{unif}}(g)\right)}{\sum_{j=1}^{m} \exp\left(\eta m \sum_{k=m+1}^{t} \gamma_k^{\text{unif}}(j)\right)}. \tag{32}$$

This completes the proof of Proposition 2.

## H  ONE-STEP MD AS A FIRST-ORDER BAYESIAN APPROXIMATION

In this section, we formally establish a theoretical connection between the one-step Mirror Descent (MD) estimator and the true Bayesian posterior mean. We demonstrate that their respective first-order Taylor expansions around a state of "no evidence" are identical up to a scalar constant. This

result provides a rigorous basis for understanding the one-step MD estimator as a principled approximation to the Bayes-optimal predictor, particularly in a low-data or low-signal regime.

The analysis hinges on treating both estimators as functions of the log-likelihood gradient evaluated at the center of the simplex, $g := \nabla_\lambda \ell(\lambda^{(0)})$, and expanding them around the point of no evidence, $g = 0$. The approximations will be expressed using Landau notation, where a vector function $f(g) = O(\|g\|^p)$ signifies that $\|f(g)\| \leq C\|g\|^p$ for some constant $C$ in a neighborhood of $g = 0$.

**Proposition 6** (First-Order Approximation of the One-Step MD Estimator). *Let $\hat{\lambda}^{MD}(g)$ be the one-step MD estimator defined as $\hat{\lambda}^{MD}(g) = \mathrm{softmax}(\eta g)$, viewed as a function of the log-likelihood gradient $g$. Its first-order Taylor expansion around $g = 0$ is given by:*

$$\hat{\lambda}_k^{MD}(g) = \frac{1}{m} + \frac{\eta}{m}(g_k - \bar{g}) + O(\|g\|^2), \tag{33}$$

*where $\bar{g} = \frac{1}{m}\sum_{j=1}^m g_j$. In vector form, this is $\hat{\lambda}^{MD}(g) = \lambda^{(0)} + \frac{\eta}{m}\mathrm{Proj}_\Delta(g) + O(\|g\|^2)$, where $\lambda^{(0)} = (1/m, \ldots, 1/m)$ and $\mathrm{Proj}_\Delta$ is the operator that projects a vector onto the hyperplane of vectors that sum to zero.*

*Proof.* The one-step MD update is given by the softmax function, $\hat{\lambda}_k^{MD}(g) = \frac{\exp(\eta g_k)}{\sum_{j=1}^m \exp(\eta g_j)}$. Since the exponential function is analytic, the softmax function is also analytic in $g$, and its Taylor series expansion around $g = 0$ exists. The expansion is given by:

$$\hat{\lambda}^{MD}(g) = \hat{\lambda}^{MD}(0) + J_{\hat{\lambda}^{MD}}(0)g + O(\|g\|^2),$$

where $J_{\hat{\lambda}^{MD}}(0)$ is the Jacobian matrix of $\hat{\lambda}^{MD}(g)$ evaluated at $g = 0$.

**Zeroth-Order Term:** At $g = 0$, the estimator evaluates to the uniform distribution, which is the prior mean $\lambda^{(0)}$:

$$\hat{\lambda}_k^{MD}(0) = \frac{\exp(0)}{\sum_{j=1}^m \exp(0)} = \frac{1}{m} = \lambda_k^{(0)}.$$

**First-Order Term (Jacobian):** The entries of the Jacobian matrix, $J_{kj}(g) = \frac{\partial}{\partial g_j}\hat{\lambda}_k^{MD}(g)$, are given by $\eta \cdot \hat{\lambda}_k^{MD}(g)(\delta_{kj} - \hat{\lambda}_j^{MD}(g))$. Evaluating at $g = 0$, where $\hat{\lambda}_j^{MD}(0) = 1/m$ for all $j$:

$$\left.\frac{\partial \hat{\lambda}_k^{MD}}{\partial g_j}\right|_{g=0} = \eta \cdot \frac{1}{m}\left(\delta_{kj} - \frac{1}{m}\right).$$

**Assembling the Expansion:** The $k$-th component of the expansion is $\hat{\lambda}_k^{MD}(g) = \hat{\lambda}_k^{MD}(0) + \sum_{j=1}^m \frac{\partial \hat{\lambda}_k^{MD}}{\partial g_j}\big|_0 \cdot g_j + O(\|g\|^2)$:

$$\hat{\lambda}_k^{MD}(g) = \frac{1}{m} + \sum_{j=1}^m \left[\frac{\eta}{m}\left(\delta_{kj} - \frac{1}{m}\right)\right]g_j + O(\|g\|^2)$$

$$= \frac{1}{m} + \frac{\eta}{m}\left(\sum_{j=1}^m \delta_{kj}g_j - \frac{1}{m}\sum_{j=1}^m g_j\right) + O(\|g\|^2)$$

$$= \frac{1}{m} + \frac{\eta}{m}(g_k - \bar{g}) + O(\|g\|^2).$$

This completes the proof. $\square$

Next, we derive the corresponding approximation for the Bayesian posterior mean. We linearize the log-likelihood around the prior mean, $\lambda^{(0)}$, which allows for an analytical treatment of the posterior.

**Proposition 7** (First-Order Approximation of the Bayesian Posterior Mean). *Consider a Bayesian model with a uniform Dirichlet(**1**) prior over $\lambda \in \Delta_{m-1}$ and a log-likelihood linearized around*

the prior mean, $\ell(\boldsymbol{\lambda}) = \ell(\boldsymbol{\lambda}^{(0)}) + \langle \boldsymbol{g}, \boldsymbol{\lambda} - \boldsymbol{\lambda}^{(0)} \rangle$. *The resulting posterior mean,* $\hat{\boldsymbol{\lambda}}^{Bayes}(\boldsymbol{g})$, *has a first-order Taylor expansion around* $\boldsymbol{g} = \boldsymbol{0}$ *given by:*

$$\hat{\lambda}_k^{Bayes}(\boldsymbol{g}) = \frac{1}{m} + \frac{1}{m(m+1)}(g_k - \bar{g}) + O(\|\boldsymbol{g}\|^2). \tag{34}$$

*In vector form, this is* $\hat{\boldsymbol{\lambda}}^{Bayes}(\boldsymbol{g}) = \boldsymbol{\lambda}^{(0)} + \mathrm{Cov}_{\boldsymbol{\lambda}^{(0)}}(\boldsymbol{\lambda})\boldsymbol{g} + O(\|\boldsymbol{g}\|^2)$, *where* $\mathrm{Cov}_{\boldsymbol{\lambda}^{(0)}}(\boldsymbol{\lambda})$ *is the co-variance matrix of the prior distribution.*

*Proof.* Under the linearized likelihood, the posterior density is $p(\boldsymbol{\lambda}|\boldsymbol{g}) \propto p(\boldsymbol{\lambda})\exp(\ell(\boldsymbol{\lambda})) \propto \exp(\langle \boldsymbol{g}, \boldsymbol{\lambda} \rangle)$, where terms constant in $\boldsymbol{\lambda}$ are absorbed into the normalization constant. The posterior mean is:

$$\hat{\boldsymbol{\lambda}}^{\mathrm{Bayes}}(\boldsymbol{g}) = \frac{\int_{\Delta_{m-1}} \boldsymbol{\lambda} \cdot \exp(\langle \boldsymbol{g}, \boldsymbol{\lambda} \rangle)\, d\mu(\boldsymbol{\lambda})}{\int_{\Delta_{m-1}} \exp(\langle \boldsymbol{g}, \boldsymbol{\lambda} \rangle)\, d\mu(\boldsymbol{\lambda})},$$

where $d\mu(\boldsymbol{\lambda})$ is the uniform probability measure over the simplex $\Delta_{m-1}$. Let $N_k(\boldsymbol{g})$ be the numerator's $k$-th component and $Z(\boldsymbol{g})$ be the denominator. The function $\hat{\boldsymbol{\lambda}}^{\mathrm{Bayes}}(\boldsymbol{g})$ is analytic, allowing a Taylor expansion.

**Zeroth-Order Term:** At $\boldsymbol{g} = \boldsymbol{0}$, the exponential term is 1. The posterior equals the prior, so the posterior mean is the prior mean:

$$\hat{\boldsymbol{\lambda}}^{\mathrm{Bayes}}(\boldsymbol{0}) = \frac{\int_{\Delta_{m-1}} \boldsymbol{\lambda}\, d\mu(\boldsymbol{\lambda})}{\int_{\Delta_{m-1}} 1\, d\mu(\boldsymbol{\lambda})} = \mathbb{E}_{\boldsymbol{\lambda} \sim \mathrm{Dir}(\boldsymbol{1})}[\boldsymbol{\lambda}] = \boldsymbol{\lambda}^{(0)}.$$

**First-Order Term (Jacobian):** The Jacobian entries are $\frac{\partial \hat{\lambda}_k}{\partial g_j} = \frac{\partial}{\partial g_j}\left(\frac{N_k}{Z}\right)$. Using the quotient rule:

$$\frac{\partial \hat{\lambda}_k}{\partial g_j} = \frac{1}{Z^2}\left(Z\frac{\partial N_k}{\partial g_j} - N_k\frac{\partial Z}{\partial g_j}\right).$$

We find the required derivatives of $N_k(\boldsymbol{g})$ and $Z(\boldsymbol{g})$ differentiating under the integral sign, which is applicable here as the integrands are continuous on the compact domain $\Delta_{m-1}$:

$$\frac{\partial Z}{\partial g_j} = \frac{\partial}{\partial g_j}\int_{\Delta_{m-1}} \exp\left(\sum_i g_i\lambda_i\right) d\mu(\boldsymbol{\lambda}) = \int_{\Delta_{m-1}} \lambda_j \exp\left(\sum_i g_i\lambda_i\right) d\mu(\boldsymbol{\lambda})$$

$$\frac{\partial N_k}{\partial g_j} = \frac{\partial}{\partial g_j}\int_{\Delta_{m-1}} \lambda_k \exp\left(\sum_i g_i\lambda_i\right) d\mu(\boldsymbol{\lambda}) = \int_{\Delta_{m-1}} \lambda_k\lambda_j \exp\left(\sum_i g_i\lambda_i\right) d\mu(\boldsymbol{\lambda})$$

Now, we evaluate these components at $\boldsymbol{g} = \boldsymbol{0}$:

- $Z(\boldsymbol{0}) = \int 1\, d\mu(\boldsymbol{\lambda}) = 1$.

- $N_k(\boldsymbol{0}) = \int \lambda_k\, d\mu(\boldsymbol{\lambda}) = \mathbb{E}[\lambda_k] = 1/m$.

- $\left.\frac{\partial Z}{\partial g_j}\right|_{\boldsymbol{g}=\boldsymbol{0}} = \int_{\Delta_{m-1}} \lambda_j \exp(\langle \boldsymbol{0}, \boldsymbol{\lambda} \rangle)\, d\mu(\boldsymbol{\lambda}) = \int \lambda_j\, d\mu(\boldsymbol{\lambda}) = \mathbb{E}[\lambda_j] = 1/m$.

- $\left.\frac{\partial N_k}{\partial g_j}\right|_{\boldsymbol{g}=\boldsymbol{0}} = \int_{\Delta_{m-1}} \lambda_k\lambda_j \exp(\langle \boldsymbol{0}, \boldsymbol{\lambda} \rangle)\, d\mu(\boldsymbol{\lambda}) = \int \lambda_k\lambda_j\, d\mu(\boldsymbol{\lambda}) = \mathbb{E}[\lambda_k\lambda_j]$.

Substituting these evaluated terms into the quotient rule expression at $\boldsymbol{g} = \boldsymbol{0}$:

$$\left.\frac{\partial \hat{\lambda}_k}{\partial g_j}\right|_{\boldsymbol{g}=\boldsymbol{0}} = \frac{1 \cdot \mathbb{E}[\lambda_k\lambda_j] - (1/m) \cdot (1/m)}{1^2} = \mathbb{E}[\lambda_k\lambda_j] - \mathbb{E}[\lambda_k]\mathbb{E}[\lambda_j] = \mathrm{Cov}_{\boldsymbol{\lambda}^{(0)}}(\lambda_k, \lambda_j).$$

This establishes that the Jacobian of the posterior mean at $\boldsymbol{g} = \boldsymbol{0}$ is the prior covariance matrix.

**Assembling the Expansion:** For a Dirichlet($\mathbf{1}$) distribution, the covariance matrix is $\text{Cov}(\lambda_k, \lambda_j) = \frac{m\delta_{kj}-1}{m^2(m+1)}$. The $k$-th component of the expansion is:

$$\hat{\lambda}_k^{\text{Bayes}}(\boldsymbol{g}) = \frac{1}{m} + \sum_{j=1}^{m} \frac{m\delta_{kj}-1}{m^2(m+1)} g_j + O(\|\boldsymbol{g}\|^2)$$

$$= \frac{1}{m} + \frac{1}{m^2(m+1)}\left(mg_k - \sum_{j=1}^{m} g_j\right) + O(\|\boldsymbol{g}\|^2)$$

$$= \frac{1}{m} + \frac{m}{m^2(m+1)}(g_k - \bar{g}) + O(\|\boldsymbol{g}\|^2)$$

$$= \frac{1}{m} + \frac{1}{m(m+1)}(g_k - \bar{g}) + O(\|\boldsymbol{g}\|^2).$$

This completes the proof of the proposition. $\qquad\square$

By comparing the results of Proposition 6 and Proposition 7, we arrive at our main result.

**Theorem 3** (First-Order Equivalence of the Estimators (restated)). *Let $\hat{\boldsymbol{\lambda}}^{MD}(\boldsymbol{g};\eta)$ be the one-step MD estimator with learning rate $\eta$, and let $\hat{\boldsymbol{\lambda}}^{Bayes}(\boldsymbol{g})$ be the Bayesian posterior mean under the linearized likelihood. There exists a unique learning rate $\eta = \frac{1}{m+1}$ for which the two estimators are first-order equivalent at $\boldsymbol{g} = \boldsymbol{0}$.*

*Proof.* Two estimators are first-order equivalent at $\boldsymbol{g} = \boldsymbol{0}$ if their values and their Jacobian matrices are identical at that point. As shown in Propositions 6 and 7, the first condition, $\hat{\boldsymbol{\lambda}}^{MD}(\boldsymbol{0};\eta) = \hat{\boldsymbol{\lambda}}^{Bayes}(\boldsymbol{0})$, holds for any $\eta$. The equivalence thus depends on matching their Jacobians.

From the proof of Proposition 6, the Jacobian of the MD estimator at $\boldsymbol{g} = \boldsymbol{0}$ is:

$$J_{\hat{\boldsymbol{\lambda}}^{MD}}(\boldsymbol{0}) = \frac{\eta}{m}\left(\boldsymbol{I} - \frac{1}{m}\mathbf{1}\mathbf{1}^T\right).$$

From the proof of Proposition 7, the Jacobian of the Bayesian posterior mean is:

$$J_{\hat{\boldsymbol{\lambda}}^{Bayes}}(\boldsymbol{0}) = \frac{1}{m(m+1)}\left(\boldsymbol{I} - \frac{1}{m}\mathbf{1}\mathbf{1}^T\right).$$

Equating the two Jacobians, $J_{\hat{\boldsymbol{\lambda}}^{MD}}(\boldsymbol{0}) = J_{\hat{\boldsymbol{\lambda}}^{Bayes}}(\boldsymbol{0})$, requires their scalar coefficients to be equal, since the matrix factor is non-zero for $m > 1$:

$$\frac{\eta}{m} = \frac{1}{m(m+1)}.$$

Solving for $\eta$ yields the unique solution $\eta = \frac{1}{m+1}$. $\qquad\square$

This theorem provides a theoretical justification for the performance of the one-step MD estimator. It demonstrates that this simple, non-iterative update is not merely a heuristic but a principled, first-order approximation of the Bayesian posterior mean under a linearized likelihood.

**Remark (The Role of $\eta$ and the Small-Gradient Assumption).** The first-order equivalence established in Theorem 1 holds in the regime where $\|\boldsymbol{g}\| \to 0$, as this is where the higher-order terms, $O(\|\boldsymbol{g}\|^2)$, are negligible. For many models, the gradient's magnitude, $\|\boldsymbol{g}\|$, scales with the amount of data (e.g., sequence length $T$), which appears to invalidate the approximation when the data size is large.

However, the one-step MD estimator, $\hat{\boldsymbol{\lambda}}^{MD} = \text{softmax}(\eta\boldsymbol{g})$, can remain a well-behaved estimator even for large $\|\boldsymbol{g}\|$ if $\eta$ is scaled appropriately. The term $\eta\boldsymbol{g}$ determines the softmax behavior. If we set the learning rate to be inversely proportional to the signal strength, for instance $\eta(T) = \Theta(1/T)$, the norm of the argument, $\|\eta\boldsymbol{g}\|$, can remain bounded. This scaling prevents the softmax output from saturating and allows the estimator to remain sensitive to the information in $\boldsymbol{g}$.

On the Bayesian side, under the linearized likelihood, the posterior mean $\hat{\boldsymbol{\lambda}}^{\text{Bayes}}(\boldsymbol{g})$ lacks a scaling parameter analogous to $\eta$. For large $\|\boldsymbol{g}\|$, the posterior density $\exp(\langle \boldsymbol{g}, \boldsymbol{\lambda} \rangle)$ concentrates sharply at the vertex of the simplex maximizing the inner product with $\boldsymbol{g}$. In this case, the first-order approximation from Proposition 7 breaks down as the neglected $O(\|\boldsymbol{g}\|^2)$ term becomes dominant.

The equivalence in Theorem 1 should therefore be interpreted as a local consistency result at $\boldsymbol{g} = \boldsymbol{0}$. It shows that for low-signal scenarios, the MD update is a principled approximation to the Bayesian one, and it provides a theoretically grounded value for $\eta$ in that regime. The empirical success of the one-step estimator for large $T$ suggests that while the functional forms of the two estimators diverge beyond the first order, a properly tuned MD estimator may still serve as an effective proxy for the Bayesian posterior mean, motivating analytical frameworks beyond local Taylor expansions.

## I  LEARNING-RATE SCALING VIA A LIPSCHITZ (SMOOTHNESS) CONSTANT

In our main analysis, we established a first-order equivalence between the one-step MD estimator and the Bayesian posterior mean in the regime of "no evidence," where the log-likelihood gradient $\boldsymbol{g} = \boldsymbol{0}$. However, for a sequence of length $T$, the magnitude of the gradient, $\|\boldsymbol{g}\|$, typically scales with $T$. This raises the question of how the learning rate of the one-step MD estimator, $\hat{\boldsymbol{\lambda}}^{\text{MD}} = \text{softmax}(\eta \boldsymbol{g})$ should be scaled with $T$ to maintain good performance.

In this section, we provide a theoretical justification for scaling the learning rate as $\eta = \Theta(1/T)$. We demonstrate that the negative log-likelihood function, viewed as a loss function over the simplex, has a gradient that is Lipschitz continuous with a constant $L$ that grows linearly with the sequence length $T$. Standard optimization theory suggests setting the learning rate inversely proportional to this Lipschitz constant, i.e., $\eta \propto 1/L$, to ensure stable updates. We use the same notation as the before:

$$c_{t,g} = \pi(y_{t-g}, y_t), \qquad t = m+1, \ldots, T, \qquad g = 1, \ldots, m,$$

and $m$ is the number of lags.

**Assumption.** We assume there exists a constant

$$c_{\min} > 0$$

such that for every $t \in \{m+1, \ldots, T\}$ and every $g \in \{1, \ldots, m\}$,

$$c_{t,g} = \pi(y_{t-g}, y_t) \geq c_{\min}. \tag{35}$$

Because each $c_{t,g}$ is a conditional probability, we also have the trivial upper bound

$$c_{t,g} \leq 1 \qquad \text{for all } t, g. \tag{36}$$

Under these assumptions the denominators that appear in derivatives are uniformly bounded away from zero, and global, uniform bounds on the Hessian are valid.

**Lemma 1** (Hessian decomposition). *For* $\ell(\boldsymbol{\lambda}) = \sum_{t=m+1}^{T} \log\left(\sum_{g=1}^{m} \lambda_g c_{t,g}\right)$ *the Hessian satisfies*

$$\nabla^2 \ell(\boldsymbol{\lambda}) = -\sum_{t=m+1}^{T} \frac{\mathbf{c}_t \mathbf{c}_t^\top}{\left(\sum_{g=1}^{m} \lambda_g c_{t,g}\right)^2}, \qquad \mathbf{c}_t := (c_{t,1}, \ldots, c_{t,m})^\top.$$

*Hence the Hessian of the negative log-likelihood* $f(\boldsymbol{\lambda}) = -\ell(\boldsymbol{\lambda})$ *is*

$$\nabla^2 f(\boldsymbol{\lambda}) = \sum_{t=m+1}^{T} \frac{\mathbf{c}_t \mathbf{c}_t^\top}{\left(\sum_{g=1}^{m} \lambda_g c_{t,g}\right)^2}.$$

*Proof.* Direct differentiation of $\ell(\boldsymbol{\lambda})$ yields the displayed formulas. □

Write $s_t(\boldsymbol{\lambda}) := \sum_{g=1}^{m} \lambda_g c_{t,g}$ and consequently the Hessian of the loss is

$$\nabla^2 f(\boldsymbol{\lambda}) = \sum_{t=m+1}^{T} \frac{\mathbf{c}_t \mathbf{c}_t^\top}{s_t(\boldsymbol{\lambda})^2}. \tag{37}$$

Each summand in equation 37 is a rank-one positive semi-definite matrix.

### I.1 UNIFORM OPERATOR-NORM BOUND ON THE HESSIAN

We now derive a uniform bound on the spectral (operator) norm of $\nabla^2 f(\boldsymbol{\lambda})$ that depends linearly on $T - m$.

**Lemma 2** (Operator-norm bound). *Under the standing assumption equation 35 (and equation 36), for every $\boldsymbol{\lambda}$ in the full simplex $\Delta_{m-1} = \{\boldsymbol{\lambda} \geq 0, \sum_g \lambda_g = 1\}$ we have*

$$\left\|\nabla^2 f(\boldsymbol{\lambda})\right\|_{\mathrm{op}} \leq (T - m)\,\frac{m}{c_{\min}^2}. \tag{38}$$

*Proof.* From equation 37 and subadditivity of the operator norm,

$$\left\|\nabla^2 f(\boldsymbol{\lambda})\right\|_{\mathrm{op}} = \Big\|\sum_{t=m+1}^{T}\frac{\mathbf{c}_t\mathbf{c}_t^\top}{s_t(\boldsymbol{\lambda})^2}\Big\|_{\mathrm{op}} \leq \sum_{t=m+1}^{T}\frac{\left\|\mathbf{c}_t\mathbf{c}_t^\top\right\|_{\mathrm{op}}}{s_t(\boldsymbol{\lambda})^2}.$$

For a rank-one matrix $\mathbf{u}\mathbf{u}^\top$ the operator norm equals $\|\mathbf{u}\|_2^2$. Hence

$$\left\|\mathbf{c}_t\mathbf{c}_t^\top\right\|_{\mathrm{op}} = \|\mathbf{c}_t\|_2^2 = \sum_{g=1}^{m} c_{t,g}^2.$$

Using equation 36 we get the upper bound $\|\mathbf{c}_t\|_2^2 \leq m \cdot 1^2 = m$ for every $t$.

For the denominator, by equation 35 and since $\sum_{g=1}^{m} \lambda_g = 1$,

$$s_t(\boldsymbol{\lambda}) = \sum_{g=1}^{m} \lambda_g c_{t,g} \geq \sum_{g=1}^{m} \lambda_g c_{\min} = c_{\min}.$$

Therefore for every $t$,

$$\frac{\left\|\mathbf{c}_t\mathbf{c}_t^\top\right\|_{\mathrm{op}}}{s_t(\boldsymbol{\lambda})^2} \leq \frac{m}{c_{\min}^2}.$$

Summing over $t = m + 1, \ldots, T$ yields

$$\left\|\nabla^2 f(\boldsymbol{\lambda})\right\|_{\mathrm{op}} \leq \sum_{t=m+1}^{T}\frac{m}{c_{\min}^2} = (T - m)\,\frac{m}{c_{\min}^2},$$

which is the bound equation 38. $\qquad\square$

### I.2 IMPROVED BOUND AT THE CENTER OF THE SIMPLEX

Here we show that evaluating exactly at the uniform vector $\boldsymbol{\lambda}^*$ yields a strictly better constant: the spectral norm of the Hessian at $\boldsymbol{\lambda}^*$ is bounded by $(T - m)\,m^2$. This gives a less pessimistic Lipschitz constant and hence a looser restriction on the conservative step-size $\eta$.

**Proposition 8** (Operator-norm bound at the uniform vector). *Let $\boldsymbol{\lambda}^* = (1/m, \ldots, 1/m)$. For the loss $f(\boldsymbol{\lambda}) = -\ell(\boldsymbol{\lambda})$ we have*

$$\left\|\nabla^2 f(\boldsymbol{\lambda}^*)\right\|_{\mathrm{op}} \leq (T - m)\,m^2. \tag{39}$$

*In particular, the gradient $\nabla f$ is Lipschitz at $\boldsymbol{\lambda}^*$ with constant $L^* \leq (T - m)m^2$, and the conservative step-size choice $\eta \leq 1/L^*$ yields $\eta = \Theta(1/T)$ for fixed $m$.*

*Proof.* Recall the Hessian decomposition (Eq. equation 37):

$$\nabla^2 f(\boldsymbol{\lambda}) = \sum_{t=m+1}^{T}\frac{\mathbf{c}_t\mathbf{c}_t^\top}{s_t(\boldsymbol{\lambda})^2}, \qquad \mathbf{c}_t = (c_{t,1}, \ldots, c_{t,m})^\top, \quad s_t(\boldsymbol{\lambda}) = \sum_{g=1}^{m} \lambda_g c_{t,g}.$$

Evaluate at the uniform vector $\boldsymbol{\lambda}^*$. Then

$$s_t(\boldsymbol{\lambda}^*) = \frac{1}{m}\sum_{g=1}^{m} c_{t,g} =: \frac{S_t}{m}, \qquad S_t := \sum_{g=1}^{m} c_{t,g}.$$

Hence the $t$-th summand becomes

$$\frac{\mathbf{c}_t \mathbf{c}_t^\top}{s_t(\boldsymbol{\lambda}^*)^2} = \frac{\mathbf{c}_t \mathbf{c}_t^\top}{(S_t/m)^2} = \frac{m^2}{S_t^2} \mathbf{c}_t \mathbf{c}_t^\top.$$

Taking operator norms and using subadditivity,

$$\left\| \nabla^2 f(\boldsymbol{\lambda}^*) \right\|_{\mathrm{op}} \le \sum_{t=m+1}^{T} \frac{m^2}{S_t^2} \left\| \mathbf{c}_t \mathbf{c}_t^\top \right\|_{\mathrm{op}}.$$

For each $t$, $\|\mathbf{c}_t \mathbf{c}_t^\top\|_{\mathrm{op}} = \|\mathbf{c}_t\|_2^2$. Observe the elementary inequality

$$\|\mathbf{c}_t\|_2^2 \le \Big( \sum_{g=1}^{m} c_{t,g} \Big)^2 = S_t^2,$$

which holds because $(\sum_i a_i)^2 = \sum_i a_i^2 + 2 \sum_{i<j} a_i a_j \ge \sum_i a_i^2$ for nonnegative $a_i$. Using this inequality we obtain, for every $t$,

$$\frac{m^2}{S_t^2} \|\mathbf{c}_t\|_2^2 \le \frac{m^2}{S_t^2} S_t^2 = m^2.$$

Summing over $t = m+1, \ldots, T$ yields

$$\left\| \nabla^2 f(\boldsymbol{\lambda}^*) \right\|_{\mathrm{op}} \le \sum_{t=m+1}^{T} m^2 = (T-m)\, m^2,$$

which proves equation 39. The remaining claims follow immediately from the standard equivalence between Hessian operator-norm bounds and local Lipschitz continuity of the gradient, and the reciprocal step-size rule $\eta \le 1/L^*$. $\qquad\square$

**Theorem 4** (Relative smoothness and $\eta$ scaling at the uniform mixture (restated))**.** *At the uniform vector $\boldsymbol{\lambda}^* = (1/m, \ldots, 1/m)$ the loss $f(\boldsymbol{\lambda}) = -\ell(\boldsymbol{\lambda})$ is $L_{rel}$-smooth relative to the KL-divergence, with*

$$L_{rel} \le (T-m)\, m^2. \tag{40}$$

*Consequently, the conservative step-size rule $\eta \le \frac{1}{L_{rel}}$ yields the asymptotic scaling*

$$\eta = \Theta\Big(\frac{1}{T}\Big) \tag{41}$$

*for fixed $m$.*

*Proof.* Proposition 8 establishes that at $\boldsymbol{\lambda}^*$ the Hessian satisfies

$$\|\nabla^2 f(\boldsymbol{\lambda}^*)\|_{\mathrm{op}} \le (T-m)\, m^2.$$

Relative smoothness of $f$ with respect to the KL-divergence means $\nabla^2 f(\boldsymbol{\lambda}) \preceq L_{\mathrm{rel}} \nabla^2 \Psi(\boldsymbol{\lambda})$, where $\Psi(\boldsymbol{\lambda}) = \sum_g \lambda_g \log \lambda_g$ is the negative entropy whose Bregman divergence is the KL. At the uniform vector, $\nabla^2 \Psi(\boldsymbol{\lambda}^*) = m\, \boldsymbol{I} \succeq \boldsymbol{I}$ (since $m \ge 1$). Therefore

$$\nabla^2 f(\boldsymbol{\lambda}^*) \preceq (T-m)\, m^2 \cdot \boldsymbol{I} \preceq (T-m)\, m^2 \cdot \nabla^2 \Psi(\boldsymbol{\lambda}^*),$$

which gives $L_{\mathrm{rel}} \le (T-m)\, m^2$ as claimed. The conservative step-size choice $\eta \le 1/L_{\mathrm{rel}}$ is therefore sufficient to guarantee stability of the mirror-descent / exponentiated-gradient update. Since $T - m = \Theta(T)$, the scaling $\eta = \Theta(1/T)$ follows for fixed $m$. $\qquad\square$

The assumption equation 35 (strict positivity of every transition probability appearing in the likelihood) is the minimal condition that guarantees a *uniform* finite Lipschitz constant $L$ over the entire simplex $\Delta_{m-1}$. If some transitions were zero, then for parameter vectors placing mass on coordinates corresponding to zero transitions some denominators $s_t(\boldsymbol{\lambda})$ could vanish and the Hessian operator norm would be unbounded (hence no global $L$ exists).

## J  ASYMPTOTIC PROPERTIES OF THE LIKELIHOOD GRADIENT

In this section, we study the asymptotic properties of the gradient of the log-likelihood function. We are mostly interested in the asymptotic scaling of the gradient with the sequence length $T$.

**Lemma 3** (Asymptotic Properties of the Gradient). *Let the true parameter $\boldsymbol{\lambda}^* \in int(\Delta_{m-1})$ induce a Markov chain on the history space $\mathcal{Y}^m$ that is aperiodic and irreducible on a finite state space. This implies the chain is geometrically ergodic. Let $\mathbb{E}_{\boldsymbol{\lambda}^*}^{stat}$ denote expectation with respect to the stationary distribution of this chain. The mean of the score vector $\mathbf{g}(\mathbf{Y})$ exhibits the following asymptotic properties as $N_{obs} \to \infty$: The expected score vector scales linearly with $N_{obs} = T - m$:*

$$\mathbf{g}_0(\boldsymbol{\lambda}^*) := \mathbb{E}_{\boldsymbol{\lambda}^*}[\mathbf{g}(\mathbf{Y})] = N_{obs} \cdot \mathbf{v}(\boldsymbol{\lambda}^*) + O(1), \tag{42}$$

*where $\mathbf{v}(\boldsymbol{\lambda}^*)$ is the constant vector of stationary expected single-step scores, whose $h$-th component is:*

$$[\mathbf{v}(\boldsymbol{\lambda}^*)]_h = m \cdot \mathbb{E}_{\boldsymbol{\lambda}^*}^{stat} \left[ \frac{\pi(Y_{t-h}, Y_t)}{\sum_{j=1}^m \pi(Y_{t-j}, Y_t)} \right]. \tag{43}$$

*The $O(1)$ term represents a constant offset due to initial conditions that does not grow with $N_{obs}$.*

*Proof.* Let us first define the score contribution from a single time step $t$ as the vector $\mathbf{z}_t(\mathbf{Y}) \in \mathbb{R}^m$, whose $h$-th component is given by:

$$[\mathbf{z}_t(\mathbf{Y})]_h = m \frac{\pi(Y_{t-h}, Y_t)}{\sum_{j=1}^m \pi(Y_{t-j}, Y_t)}.$$

The total score vector is the sum of these contributions over the observation period:

$$\mathbf{g}(\mathbf{Y}) = \sum_{t=m+1}^T \mathbf{z}_t(\mathbf{Y}), \quad \text{where } N_{\text{obs}} = T - m.$$

The process $(\mathbf{z}_t)_{t>m}$ is a sequence of random vectors. Because it is a function of the underlying ergodic Markov chain $(Y_{t-m}, \ldots, Y_t)$, the sequence $(\mathbf{z}_t)$ is also ergodic and its distribution converges to a stationary distribution. By the linearity of expectation, the expected score is the sum of the individual expectations:

$$\mathbb{E}_{\boldsymbol{\lambda}^*}[\mathbf{g}(\mathbf{Y})] = \sum_{t=m+1}^T \mathbb{E}_{\boldsymbol{\lambda}^*}[\mathbf{z}_t(\mathbf{Y})].$$

The key assumption is the geometric ergodicity of the Markov chain. This implies that the distribution of the state $(Y_{t-m}, \ldots, Y_t)$ converges exponentially fast to the unique stationary distribution, regardless of the initial state $(Y_1, \ldots, Y_m)$. Consequently, the expectation $\mathbb{E}_{\boldsymbol{\lambda}^*}[\mathbf{z}_t(\mathbf{Y})]$ converges exponentially fast to its stationary-state expectation, $\mathbf{v}(\boldsymbol{\lambda}^*)$. This convergence can be quantified. There exist a constant vector $\mathbf{C}$ and a rate $\rho \in (0, 1)$ such that for all $t > m$:

$$\|\mathbb{E}_{\boldsymbol{\lambda}^*}[\mathbf{z}_t(\mathbf{Y})] - \mathbf{v}(\boldsymbol{\lambda}^*)\| \leq \|\mathbf{C}\|\rho^{t-m}.$$

We can now rewrite the sum of expectations:

$$\mathbb{E}_{\boldsymbol{\lambda}^*}[\mathbf{g}(\mathbf{Y})] = \sum_{t=m+1}^T \left( \mathbf{v}(\boldsymbol{\lambda}^*) + (\mathbb{E}_{\boldsymbol{\lambda}^*}[\mathbf{z}_t(\mathbf{Y})] - \mathbf{v}(\boldsymbol{\lambda}^*)) \right)$$

$$= \left( \sum_{t=m+1}^T \mathbf{v}(\boldsymbol{\lambda}^*) \right) + \left( \sum_{t=m+1}^T (\mathbb{E}_{\boldsymbol{\lambda}^*}[\mathbf{z}_t(\mathbf{Y})] - \mathbf{v}(\boldsymbol{\lambda}^*)) \right)$$

$$= N_{\text{obs}} \cdot \mathbf{v}(\boldsymbol{\lambda}^*) + \mathbf{E}_T,$$

where $\mathbf{E}_T$ is the cumulative error term due to the process not having reached stationarity at early time steps. We can bound the norm of this error term:

$$\|\mathbf{E}_T\| = \left\| \sum_{t=m+1}^T (\mathbb{E}_{\boldsymbol{\lambda}^*}[\mathbf{z}_t] - \mathbf{v}(\boldsymbol{\lambda}^*)) \right\| \leq \sum_{t=m+1}^T \|\mathbb{E}_{\boldsymbol{\lambda}^*}[\mathbf{z}_t] - \mathbf{v}(\boldsymbol{\lambda}^*)\| \leq \sum_{t=m+1}^T \|\mathbf{C}\|\rho^{t-m}.$$

Let $k = t - m$. The sum becomes a geometric series:

$$\|\mathbf{E}_T\| \leq \|\mathbf{C}\| \sum_{k=1}^{T-m} \rho^k < \|\mathbf{C}\| \sum_{k=1}^{\infty} \rho^k = \|\mathbf{C}\| \frac{\rho}{1-\rho}.$$

The sum of the error terms is bounded by a finite constant that does not depend on $T$ or $N_{\text{obs}}$. Therefore, the error term is $O(1)$. This proves the first claim: $\mathbf{g}_0(\boldsymbol{\lambda}^*) = N_{\text{obs}} \cdot \mathbf{v}(\boldsymbol{\lambda}^*) + O(1)$. Note that $O(1)$ is also $o(N_{\text{obs}})$, so the term is asymptotically sub-linear. $\qquad\square$

