# OpenReview forum: "Transformers Learn Latent Mixture Models In-Context via Mirror Descent"
_ICLR.cc/2026/Conference — ICLR 2026 Poster_

### Official Review · Reviewer_y3Go · 2025-10-15

**Soundness:** 2
**Presentation:** 4
**Contribution:** 3
**Rating:** 6
**Confidence:** 3

**Summary:**

The authors set up a synthetic task where to perform well at each step the transformer must estimate mixing weights for a set of transition distributions (the mixture of transition distributions task). They show that posterior mean inference for this task is intractable and then discuss one step of mirror descent as an attractive estimator of the mixing weights. They then show theoretically and empirically that disentangled transformers implement mirror descent to learn mixture weights for this process.

**Strengths:**

1. I really like the matrix diagrams in section 4, it made following that section a lot easier. I felt that the paper was well written and structured.
2. To my knowledge no one has previously proposed that ICL involves transformers performing mirror descent. This gives a new perspective on ICL
3. I had not known of the MTD model before reading this, it appears to be a fairly well motivated synthetic task in this setting.

**Weaknesses:**

The lack of any experiments involving non-disentangled transformers weakens this papers claims. It's not clear to me that a general non-disentangled transformer would implement this mirror descent step. If you consider the Nichani et al. (2024) paper cited in this work, they also perform a theoretical and empirical analysis with disentangled transformers, but they include experiments involving standard non-disentangled transformers also. Something similar in this case would greatly improve the strength of this papers evidence for it's claims and make the title more justified.

**Questions:**

- Is there any reason why 3 and 6 (or 5, there appears to be an inconsistency between the main text and the appendix) layers were chosen? Does it also hold for other layer numbers?

- Are there any plans to extend this to a more realistic data generating process where tokens can depend on multiple previous tokens at different time lags?

- Would the 5/6 layer disentangled transformer be amenable to the same sort of analysis as the three layer model, showing that it implements two step MD?

---

> ### Author Response · Authors · 2025-12-01
> **Response**
>
> We thank the reviewer for appreciating our work, and we are glad to know that the matrix diagrams were useful for understanding the paper.
>
> We respond to the reviewer's questions as follows:
>
> **Standard transformer experiments** In the revised manuscript, we added experiments with a standard transformer including learned embeddings (compared to one-hot), query–key–value projection,s attention heads, residual addition (instead of concatenation) (see Section Experiments and Appendix Experimental Details). This model achieves comparable KL performance to the disentangled transformer and shows matching attention patterns across the three layers, which strengthens our claim that the MD‑like mechanism is not an artifact of disentangling.
>
>  **Number of layers.** We appreciate the reviewer for spotting the inconsistency; the correct number in the multi‑step experiment is 5 layers, and we have fixed this in the text. The choice of 3 layers in our construction is dictated by the algorithm: the responsibilities, summation, and weighting steps naturally require three distinct rounds of attention/aggregation, and we empirically observed that 2‑layer transformers perform significantly worse on the task. For 5 layers, our intuition is that one can reuse the responsibility computed in the first MD step so that not all three stages need to be replicated to approximate a second step, but we do not yet have a clean mechanistic decomposition of the 5‑layer model, and we therefore refrain from claiming a full two‑step MD construction and leave this analysis for future work.
>
> **More realistic data‑generating processes.** We emphasize that, even in our current MTD setting with a single latent lag variable, the Bayes‑optimal predictive distribution already depends on *all* past tokens through a convex combination of lag‑specific transitions, $ \sum_g \hat\lambda_g \pi(y_{t-g}, \cdot) $, and our MD estimator approximates this multi‑lag dependence. That said, extending the framework to more complex, richer processes is an interesting direction: for instance, one could allow distinct transition matrices $ \pi^{(g)} $ for different lags or introduce context‑dependent mixture weights $ \lambda_t $, which would likely require additional capacity (e.g., MLP layers) to store and manipulate multiple transition operators, making a precise mechanistic decomposition substantially harder. We view systematically analyzing such models as important future work beyond the scope of this paper, whose goal is to first pin down the MD mechanism in a stationary MTD setting.

---

### Official Review · Reviewer_wFmK · 2025-10-20

**Soundness:** 3
**Presentation:** 3
**Contribution:** 2
**Rating:** 4
**Confidence:** 3

**Summary:**

This paper introduces a framework for in-context learning with latent variable modelling, based on a mixture of transition distributions.
The authors introduce a synthetic task that reframes the estimation of past tokens to learning latent mixture weights. The paper also proves that the learned algorithm is mirror descent. The paper also empirically shows that the learned algorithm for a **disentangled** transformer is mirror descent. Finally, the paper shows that the estimator is an approximation of the Bayesian optimal solution.

**Strengths:**

The paper has the following strengths:
1. Clear framework presentation with latent variable modelling.
2. Good theory for the disentangled transformer
3. Good link to Bayse optimal solution

**Weaknesses:**

1. While the theory is solid and I consider it a good contribution, the main problem is that the results are focused only on the disentangled transformer.
 - The paper claims that the results were proved for the general transformer (for example, in lines 480-481).
-  The authors should make it clear that they refer to a disentangled transformer, which is not the same as the generally used one for practical tasks.
2. The paper should try to extend results to the general case transformer, or state why they do not work.
3. The experimental section should be improved.
- Could more experiments be added for real tasks like language modelling?
- Could the models in experiments be extended to the general transformer?
4. Small typos:
- 076, "patterns math our"?
- 205-206: "by a Transformer" Do you mean a disentangled transformer?

**Questions:**

Could more experiments be added for real tasks like language modelling?
Could the models in experiments be extended to the general transformer?

---

> ### Author Response · Authors · 2025-12-01
> **Response**
>
> We thank the reviewer for their comments, and we are glad they appreciated our work. Here is our response to the comments , which we hope will help address the reviewer's questions and concerns.
>
> **Standard Transformer New Experiments**
> We agree that our formal construction is stated for the disentangled transformer. In the revised manuscript, we now make this explicit whenever we refer to the theoretical results (e.g., in the model and construction sections). At the same time, we added new experiments with a standard transformer including learned embeddings, query–key–value projections, residual addition instead of concatenation, (Appendix Hyperparameters, $ \mathcal{T}_{\text{standard}} $). These models achieve comparable KL performance and show matching attention patterns across the three layers, indicating that our conclusions are not an artifact of the disentangled architecture. We think that these experiments improve the strength of our claims.
>
> **Why we do not have language‑modeling experiments.**
> While we agree that testing these ideas on large language models and real text would be very interesting, it is beyond the scope of the present work. Our goal is not to benchmark on text nor to claim that MTD is a good model for text, but to *identify the algorithm* learned in a controlled setting where ground‑truth components (MTD, Bayes, MD) are known. This mirrors much of the ICL/mechanistic literature, which also studies synthetic regression or Markov tasks rather than natural language e.g., gradient‑descent ICL in linear regression and related settings [2-9] and sequential models / Markov chains [10–13] (note how they also do not have any text experiments in their works). Our paper is fundamentally theoretical: we deliberately work in a controlled MTD setting where the data‑generating process is known, so that the Bayes predictor and the corresponding Mirror Descent updates are precisely defined and can be checked against the trained transformer. For real‑world text, the likelihood model is unknown, so it is not clear what the “ground‑truth” MD updates would be, nor how one would rigorously verify that a large language model is implementing them rather than some other estimator. We therefore believe that a careful analysis on a synthetic but nontrivial MTD task, together with the added standard‑transformer experiments, is the appropriate scope for this paper. It is certainly interesting to ask whether analogous MD‑like mechanisms emerge in large language models trained on text, but answering that question would require a separate, substantially larger empirical study, which we leave for future work.
>
> **Typos:** Thank you for the typos! We fixed them in the updated manuscript.

---

> > ### Author Response · Authors · 2025-12-01
> >
> > [2] Garg, Shivam, Dimitris Tsipras, Percy S. Liang, and Gregory Valiant. “What can transformers learn in-context? A case study of simple function classes.” Advances in neural information processing systems 35, 2022: 30583–30598.
> >
> > [3] Akyürek, Ekin, Dale Schuurmans, Jacob Andreas, Tengyu Ma, and Denny Zhou. “What learning algorithm is in-context learning? Investigations with linear models.” In The Eleventh International Conference on Learning Representations (ICLR), 2023.
> >
> > [4] Bai, Yu, Fan Chen, Huan Wang, Caiming Xiong, and Song Mei. “Transformers as statisticians: Provable in-context learning with in-context algorithm selection.” Advances in neural information processing systems 36, 2023.
> >
> > [5] Von Oswald, Johannes, Eyvind Niklasson, Ettore Randazzo, João Sacramento, Alexander Mordvintsev, Andrey Zhmoginov, and Max Vladymyrov. “Transformers learn in-context by gradient descent.” In International Conference on Machine Learning (ICML), 2023: 35151–35174.
> >
> > [6] Von Oswald, Johannes, Eyvind Niklasson, Maximilian Schlegel, Seijin Kobayashi, Nicolas Zucchet, Nino Scherrer, Nolan Miller, Mark Sandler, Max Vladymyrov, and Razvan Pascanu. “Uncovering mesa-optimization algorithms in transformers.” arXiv preprint arXiv:2309.05858, 2023.
> >
> > [7] Ahn, Kwangjun, Xiang Cheng, Hadi Daneshmand, and Suvrit Sra. “Transformers learn to implement preconditioned gradient descent for in-context learning.” Advances in neural information processing systems 36, 2023.
> >
> > [8] Fu, Deqing, Tian-Qi Chen, Robin Jia, and Vatsal Sharan. “Transformers Learn to Achieve Second-Order Convergence Rates for In-Context Linear Regression.” In Advances in Neural Information Processing Systems 37, 2024.
> >
> > [9] Zhang, Ruiqi, Spencer Frei, and Peter L. Bartlett. Trained transformers learn linear models in-context.” Journal of Machine Learning Research 25, no. 49 (2024): 1–55.
> >
> > [10] Nichani, Eshaan, Alex Damian, and Jason D. Lee. “How Transformers Learn Causal Structure with Gradient Descent.” In Forty-first International Conference on Machine Learning (ICML), 2024.
> >
> > [11] Edelman, Ezra, Nikolaos Tsilivis, Benjamin L. Edelman, Eran Malach, and Surbhi Goel. “The evolution of statistical induction heads: In-context learning Markov chains.” Advances in neural information processing systems 37 (2024): 64273–64311.
> >
> > [12] Bietti, Alberto, Vivien Cabannes, Diane Bouchacourt, Hervé Jégou, and Léon Bottou. “Birth of a transformer: A memory viewpoint.” Advances in neural information processing systems 36 (2023).
> >
> > [13] Chen, Siyu, Heejune Sheen, Tianhao Wang, and Zhuoran Yang. “Unveiling Induction Heads: Provable Training Dynamics and Feature Learning in Transformers.” In The Thirty-eighth Annual Conference on Neural Information Processing Systems 37, 2024.

---

### Official Review · Reviewer_vGtX · 2025-10-28

**Soundness:** 3
**Presentation:** 4
**Contribution:** 3
**Rating:** 8
**Confidence:** 3

**Summary:**

This paper investigates the Mixture of Transition Distribution (MTD) task, where a latent variable must be learned in-context. The authors show that an ad hoc disentangled transformer is able to perform one step of Mirror Descent (MD), a mechanism derived from initializing on a specific prior distribution. This MD step serves as a first-order approximation of the Bayesian posterior mean. For longer sequences, they empirically demonstrate that Multi-step Mirror Descent approximates the Bayesian posterior mean more effectively. They also show that training disentangled transformers match the results obtained by their construction.

**Strengths:**

The primary strength of this paper lies in its conceptual originality and technical rigor in tackling the challenge of dynamic latent variable inference. The paper's most original step is presenting the Mixture of Transition Distributions (MTD) model as an in-context learning task requiring the dynamic estimation of latent mixture weights. This approach successfully moves the field past simpler models (like linear regression or elementary Markov Chains). By identifying and proving that a disentangled transformer implements one step of the MD algorithm, the paper establishes a new insight: a trained disentangled transformer (or the attention part of transformers) in this context can effectively approximate the Bayesian posterior mean and the natural algorithm to achieve it. This provides an algorithmic explanation for how transformers update their beliefs about dynamic sequence properties, representing a good contribution to understanding transformer capacity.

**Weaknesses:**

I believe there might be a limitation in the scope of the generalization claim. To truly support the title's broad assertion that transformers learn via Mirror Descent, the paper should include an experiment demonstrating that this mechanism holds up in a standard transformer architecture (including MLPs). Furthermore, since the formal proof only covers a single MD step, it would be highly valuable to provide an explicit discussion or intuition regarding the theoretical path for two or more MD steps and whether an extra iteration naturally leads to a quadratic approximation of the Bayesian posterior mean.

**Questions:**

1. On line 75 you claim the training of the transformer is using Gradient Descent, then you actually use ADAM and do not give a reason, I do not think they will both converge to the same critical point, could you comment on this?.
2. Could you give a discussion where having a known transition matrix \pi is realistic? I might not get why you were using rows from a Dirichlet distribution.
3. How might non-linear activation functions (like ReLU or GELU) interplay with the smooth, exponentiated gradient algorithm required for the MD update?

---

> ### Author Response · Authors · 2025-12-01
> **Response**
>
> We thank the reviewer for their suggestions and we are glad they appreciated our works. Here is our response to the comments:
>
> **New experiments Standard Transformer** We added new experiments where we train standard transformers with attention parametrized as query-key-value and projection for each layer and learning both the semantic and RPE embeddings, the output of each layer is now added instead of concatenated as per standard transformer architecture. The results are reported in the updated manuscript in Figure 4. The performance of the standard transformer is comparable to the disentangled transformer, and more importantly, the attention patters match both the construction and training of the disentangled transformer.
>
>
> **Multi‑step MD and theory.**
> We agree that our formal construction currently covers only a single MD step. We now make this limitation explicit in the main text. Extending our construction beyond a single MD step is non‑trivial because MD updates are multiplicative, whereas gradient descent uses additive updates that naturally arise from residual/skip connections in standard transformers. In GD, each layer can simply add a learned approximation of the gradient to the previous iterate, so stacking layers directly mimics multiple GD steps. By contrast, MD rescales the previous weights by an exponential of the responsibilities, and from the second step onward these responsibilities must be recomputed at the current iterate. As a result, implementing multiple MD steps would require the transformer to realize a sequence of operations, which interact in a much more complex way with our RPE-based architecture than the linear, additive GD updates. Empirically, Fig. 5 shows that a 5‑layer transformer matches the performance of a 2‑step MD estimator across sequence lengths. We now state this more cautiously as “the trained transformer matches the performance of 2‑step MD” and leave a precise multi‑step construction as an interesting direction for future work.
>
> **Adam vs. gradient descent.**
> We thank the reviewer for catching this ambiguity. In the revised manuscript we now state explicitly that we use Adam in the experiments; our earlier wording (“gradient descent”) was intended in the loose sense of gradient‑based optimization, not to claim we use vanilla SGD. We chose Adam for stability and faster convergence on long sequences, and our theoretical construction and MD interpretation do not depend on the specific optimizer. We also ran SGD variants and observed very similar attention patterns and KL curves, albeit with longer training times.
>
> **Known transition matrix** We believe that assuming a known transition matrix actually makes our setting *more* realistic than many prior ICL setups. In real LLMs, part of the structure of the data is plausibly stored in the weights during pretraining and then reused for in‑context prediction; our MTD task is designed to capture exactly this by letting $ \pi^\star $ be learned in‑weight while the mixture weights $ \lambda $ inferred in‑context. For text, fully memorizing long-context n-grams would require sample complexity exponential in the context length, but it is reasonable to expect that models instead learn lower-order transition structure in their weights and then adapt mixture weights based on the prompt. This is precisely the picture illustrated in Fig. 1 and clarified in the revised “Why MTD?” paragraph of the introduction.
>
> **Nonlinearities** In our construction we deliberately work with attention‑only layers, so the MD update is implemented directly by the softmax and does not rely on additional nonlinearities. In a standard transformer with ReLU/GELU MLPs, these activations could in principle learn to approximate corrections to the MD step or compose with it in more complex ways, but analyzing that interaction rigorously is beyond the scope of this work and we leave it as an interesting direction for future research.

---

### Official Review · Reviewer_cRmd · 2025-10-31

**Soundness:** 3
**Presentation:** 4
**Contribution:** 3
**Rating:** 8
**Confidence:** 4

**Summary:**

This paper studies transformers trained on the MTD (Mixture of Transition Distributions) task. They show that a three layer disentangled transformer can naturally represent one step of mirror descent on the maximum likelihood objective. They also show that trained transformers have the same attention maps as their theoretical construction and obtain similar performance to mirror descent with optimal step sizes, supporting the hypothesis that transformers solve this task via mirror descent.

**Strengths:**

The task is a natural generalization of previous in-context learning tasks studied in the literature and removes the dependence on a fixed causal graph. The paper is also very well written, especially section 4 (the explicit construction). In addition, while the experiments are relatively limited, they do support the paper's thesis that transformers solve this ICL task via mirror descent.

**Weaknesses:**

While the paper provides a construction for a three layer disentangled transformer that can express one step of mirror descent, it doesn't study optimization so it's still unclear how gradient descent finds this solution. To support the claim that transformers learn this construction, the paper relies on experiments on disentangled transformers which show that both the attention maps and the overall performance match their explicit construction. However, there are a few weaknesses of the experiments:

- They don't verify agreement in weight space between the trained transformer and their construction, they only verify the attention maps/KL divergence. Since the attention maps are input dependent, it's difficult to use a single attention map to verify equivalence of two architectures.
- The agreement in attention maps isn't very convincing, for example while the first attention map matches the diagonal structure of the explicit construction, the actual weights appear to be quite far apart. Additionally, the middle attention map has a strong banded structure which is not captured by the explicit construction.
- I believe that the paper only empirically validates their claim on the disentangled transformer. However, if they only verify final performance and attention maps, it would be good to include a "standard" transformer baseline with projections and MLPs, since you can still easily verify attention maps and performance on these. The main advantage of training a disentangled transformer is to look for weight-space agreement.
- The only evidence that transformers perform multi-step mirror descent are the performance plots in Figure 5. It would be good to verify attention maps or ideally the weights themselves to check whether this is actually true or whether it's implementing a different estimator with similar performance (although this seems unlikely).

Minor points:
- In the first part of the abstract/paper, the claim is phrased as "transformers implement Mirror Descent," however the real claim in these sections is that transformers *can* implement mirror descent, and then the empirical part of the paper tries to argue that they do.
- line 151+152: While predicting $\lambda$ is probably a necessary step for most good estimators, the actual task is just to predict $y_t$ given $y_{1:t-1}$, not to predict $\lambda$?
- line 227: overhang in the main paper

**Questions:**

- What do the weights of the trained disentangled transformer look like?
- How would the performance of the disentangled transformer compare to a standard transformer on this task? Is it clear that the MLPs won't help?
- For the multi-step transformer, were the step sizes for each step tuned independently or were they tied together? In the trained transformer can you read off the step sizes? Are they the same or does it use a schedule?

---

> ### Author Response · Authors · 2025-12-01
> **Response**
>
> We thank the reviewer for their comments and questions. Here is our response to the comments, which we hope will help address the reviewer's concerns.
>
> **Optimization** We agree that our work is mostly about the representation of MD rather than optimization. Studying optimization is notoriously difficult with multi-layer transformers. Our setup requires 3 layers to be able to represent the one-step MD estimator. We believe that even if the optimization part is mostly left to empirical observations, the representation of the MD estimator is a valuable contribution to the mechanistic ICL literature.
>
> **Weight space agreement** We believe the main advantage of the disentangled transformer is that it lets us interpret layer‑wise computations by forcing each layer’s output into a fixed, explicit subspace of the embedding space; this is what allowed us to recognize that the learned algorithm is MD. To better support agreement between the construction and the trained model, we added in Figure 4 and Appendix D heatmaps of the first-layer attention softmax $ \text{Softmax}(W_1^A) $ compared to the true transitions matrix $ \pi^\star $ for a trained model, together with the row‑wise KL divergence. We focus on attention maps rather than raw weights because, especially with three learned attention matrices and RPE tables, there are many weight‑space symmetries (e.g., reparameterizations before softmax) that implement exactly the same attention patterns and function; therefore, matching weights seems neither necessary nor robust as a criterion. For similar reasons, we only show the first layer: our construction uses for simplicity, a one‑hot encoding of the relevant relative positions, whereas the learned RPE tables need not match this encoding uniquely; we observe that they recover the same effective responsibilities but may store them differently across the embedding, making deeper layer weight comparisons hard to interpret.
>
> Regarding the banded structure in the middle attention map, we believe that this is a consequence of the fact that we have the loss computed only over the last token, therefore it is only important that the attention is uniform for the last row of the attention for the second layer.
>
> **New experiments Standard Transformer** We added new experiments where we train standard transformers with attention parametrized as query-key-value and projection for each layer and learning both the semantic and RPE embeddings, the output of each layer is now added instead of concatenated as per standard transformer architecture. The results are reported in the updated manuscript in Figure 4. The performance of the standard transformer is comparable to the disentangled transformer, and even more so, the attention patterns match both the construction and training of the disentangled transformer.
>
> **Scope clarification** We thank the reviewer for highlighting this ambiguity; in the revised manuscript, we now state explicitly that our goal is to show that transformers can learn to implement Mirror Descent on this task, rather than to claim that they necessarily converge to the MD estimator in general.
>
> **In‑context task definition** We appreciate this clarification and agree that, at the end of the day, the task is next‑token prediction given the past. Our formal task definition is meant to isolate the in‑context learning component: since the transition matrix is shared across sequences and can in principle be learned in‑weight during pretraining, the remaining quantity that must be adapted from a single sequence is the mixture weights $ \lambda $. We have clarified this perspective in the revised manuscript.

---

> ### Author Response · Authors · 2025-12-01
>
> **MLP** We agree that omitting MLPs is a limitation, and we do not think that they could not improve performance. In fact, when the attention stack implements a non‑optimal estimator (such as one‑step MD), additional MLP layers can move the model closer to the Bayes predictor (for example, by correcting the fact that we always initialize MD at the center of the simplex). Our goal in this work, however, is to identify the algorithm implemented by attention, and MLPs tend to approximate such corrections in a highly entangled, non‑interpretable way that makes the underlying mechanism harder to understand. This is why, in line with much of the mechanistic ICL literature, our main construction uses attention‑only architectures; they sacrifice some expressivity, but are crucial for obtaining a clean mechanism, which we then see re‑emerge qualitatively even in more general (standard) transformer experiments and that hopefully can give some insights on how it could be implemented in more general settings.
>
> **Multi-step MD** For the multi‑step MD baselines, we tuned the step sizes for each step independently, since we expect this to be closer to what a trained transformer can implement in practice; however, we did not observe substantial differences when we tied the step sizes, so our conclusions are not sensitive to this choice. In the trained 5‑layer transformer, we do not attempt to read off explicit step sizes: at present, we do not yet have a precise mechanistic decomposition of how multiple MD‑like steps are implemented across layers,and  extracting an effective learning rate from the weights remains an interesting direction for future investigations.

---

### Official Review · Reviewer_9cwh · 2025-10-31

**Soundness:** 3
**Presentation:** 3
**Contribution:** 2
**Rating:** 6
**Confidence:** 3

**Summary:**

The main problem the paper tackles is: How do Transformers perform in-context learning when the task requires inferring latent mixture weights on the fly instead of machine learning algorithms? To study this problem, the authors introduce a synthetic ICL task based on the Mixture of Transition Distributions (MTD) model. Methodologically, they explicitly construct a 3-layer disentangled Transformer that mirrors the one-step mirror descent algorithm. Empirically, the show that the trained transformer can approximate the theoretical construction.

**Strengths:**

* Extends the “transformers-as-optimizers” view from GD on fully observed tasks to mirror descent (exponentiated gradient) on latent mixture weights, a novel setting (latent variables over the simplex) rather than another linear-regression variant.
* The paper is well-written and easy to understand. The proof idea illustration in the main text is very helpful.
* The theoretical construction is backed up by the empirical results.

**Weaknesses:**

1. The theoretical analysis relies on a simplified, MLP-free Transformer architecture. Hence, it remains uncertain whether the same mirror-descent behavior holds in standard Transformers with MLPs (But I understand this is always the limitation of all constructive proof for almost all ICL theory papers)

2. Proposition 3 and the constructed model require access to the transition matrix $\pi*$, which limits applicability when base dynamics are not shared/known. This $\pi*$ is later encoded in the output matrix $\tilde{W}_O$. Doesn't this mean that the transformer has no chance to learn the transition matrix of the MTD model? If this is the case, how can the theoretical result justify that the transformer is "learning the latent mixture model in context" if it only learn the mixture weight, not the transition matrix? I'm not an expert in MTD, so I might misunderstand the concept, and will appreciate further explanation on this

**Questions:**

1. Why did the author choose the disentangled transformer architecture to analyze? What will happen if we add back the MLP layer?

2. I think it would be helpful if the authors could elaborate on the choice of the MTD model as a tool to analyze the latent structure of the data. The authors already provide a formal introduction in Section 3, but it might strengthen the paper if that formal definition were explicitly linked back to the intuitive example in Figure 1, which clearly motivates why we want to analyze ICL beyond the machine-learning algorithms studied in previous works.

---

> ### Author Response · Authors · 2025-12-01
> **Response**
>
> We appreciate the reviewer's comments and suggestions. Here is our response, which we hope will help clarify our work and address the reviewer's concerns.
>
> **MLP:** We acknowledge the fact that the lack of MLP is a limitation of our work, but we believe, as also pointed out by the reviewer, that this is a general limitation of most of the works about transformer and trying to understand what algorithms they implement for ICL. Our main goal is to identify the algorithm implemented by the attention stack, not to achieve the best possible performance. When the estimator implemented by attention (in our case, one‑step MD) is not Bayes‑optimal, adding MLPs will typically help to better approximate the Bayesian predictor, but in a non-interpretable way that makes the underlying mechanism difficult to understand. This is precisely why much of the mechanistic ICL literature also works with architectures without MLPs: removing them sacrifices some expressivity, but is essential for obtaining a clean, interpretable characterization of the learned mechanism, which hopefully can be then found also in more general architectures.
>
> **Why MTD** The reviewer is correct that our construction assumes access to the true transition matrix $ \pi^\star $. We chose the MTD model precisely because it lets us separate **in‑weight** and **in‑context** learning in a way that we believe is closer to how large language models operate. Much of the existing ICL literature (e.g., works that explain ICL as gradient descent for linear regression or as counting estimators for simple Markov chains) designs tasks where most of the useful structure must be inferred from the prompt; the data distribution itself plays little role beyond providing examples. In contrast, our setting explicitly models the idea that some statistical properties, here the transition matrix $ \pi^\star $, can be learned during pretraining and stored in the weights to be reused at inference time, while some other components, the mixture weights $ \lambda $, are inferred and adapted in‑context from a single given prompt. Thinking about text, it is unrealistic for a model to memorize full high‑order n-gram tables due to the exponential sample complexity, but it is plausible that they learn lower‑order transition structure in their weights and then dynamically reweights different lags depending on the context. The MTD model is a natural choice for this task and the main idea we tried to convey in the example in Figure 1. We made it more clear in the updated manuscrip. This is ideas is also reminiscent of the rule selection in [1] but we left investigating this possibility for future work.
>
> To the best of our knowledge, this is the first in‑context learning setup that explicitly couples an in‑weight component (learning \$ \pi^\star $ with an in‑context component (inferring $ \lambda $), making it a natural and realistic testbed for studying the fundamental in‑weight / in‑context dichotomy observed in LLMs.
>
> We also considered a setup where both the transition matrix pi and the mixture weights are learned in-context. However, this increases the complexity of the task and the number of layers required to have good performance on the task. We still believe that this is a valid approach and we will investigate it in future work.
>
> **New experiments learning $ \pi^\star $** To further validate that the first attention layer learns the transition matrix we added heatmaps and row-wise KL divergence between the attention matrix and the transition matrix. The results are reported in the updated manuscript in Figure 4 and in Appendix D.
>
> **Disentangled transformer** Using the disentangled transformer is simply for convenience. When training transformers with learned embeddings which are summed instead of concatenated the information is stored and communicated across layers in different subspaces of the embedding space. This makes interpreting the computation within the model difficult. The combination of disentanglement and one-hot encoding simplifies this a lot by explicitly keeping the information of each layer in a fixed subspace of the embedding. This ultimately allowed us to understand that the implemented algorithm was MD.
>
> **New Experiments Standard Transformer** To further validate that our claim is not a biproduct of the disentangled transformer we also trained a standard transformer with attention parameterized by key-query-value and projections, learned semantic and RPE embeddings and summation instead of concatenation. The results are reported in the updated manuscript in Figure 4. The performance of the standard transformer is comparable to the disentangled transformer and therefore the MD estimator, and even more so, the attention patters match both the construction and training of the disentangled transformer.
>
> [1] Nguyen, Timothy. "Understanding transformers via n-gram statistics." Advances in neural information processing systems 37 (2024): 98049-98082.

---

### Author Response · Authors · 2025-12-01
**General response**

We are grateful to all reviewers for their careful assessments and constructive suggestions. We are encouraged that the reviewers viewed our work as a conceptually original extension of the “transformers‑as‑optimizers” perspective, moving from gradient descent on regression tasks to mirror descent on latent mixture weights in a Mixture of Transition Distributions (MTD) model.

In response to the reviewers’ suggestions, we have revised the manuscript and added the following experiments, analyses, and clarifications:

- **Standard transformer experiments**:
  We added new experiments with a standard transformer (learned embeddings, query–key–value projections, residual addition instead of concatenation) and show that it achieves comparable KL performance to the disentangled transformer, with attention patterns closely matching both our construction and the disentangled model (Fig. 4).

- **Additional experiments for orders 3 and 5 MTD**:
  We included new experiments in Appendix D showing KL performance and attention patterns for MTD models of orders \(m=3\) and \(m=5\), further validating that the learned transformers track the MD estimator.

- **Attention–transition matrix alignment**:
  We added heatmaps and row‑wise KL divergence comparing the first‑layer attention matrix softmax to the true transition matrix $ \pi^\star $, further supporting that the model learns the underlying transition structure in the first layer and that the attention stack implements the intended MD‑like update (Fig. 4, Appendix D).

- **Clarifications on the MTD setting and known transition matrix**:
  We added the “Why MTD?” discussion, explaining how assuming a known (or in‑weight‑learned) transition matrix $ \pi^\star $ and inferring mixture weights $ \lambda $ in‑context mirrors the in‑weight / in‑context learning combination in LLMs, and why this provides a natural and realistic testbed for mechanistic ICL analysis.

- **Minor edits and fixes**:
  We corrected typos and improved the exposition for better readability.

We hope these additions and clarifications address the main concerns raised by the reviewers and further strengthen the paper, and we would be happy to clarify any remaining questions.

---

### Meta-Review · Area_Chair_5JJB · 2025-12-31

**Summary:**

This paper studies transformers trained on the MTD (Mixture of Transition Distributions) task. They show that a three layer disentangled transformer can naturally represent one step of mirror descent on the maximum likelihood objective. They also show that trained transformers have the same attention maps as their theoretical construction and obtain similar performance to mirror descent with optimal step sizes, supporting the hypothesis that transformers solve this task via mirror descent. All the reviewers admit the value of theoretical results presented in the paper.

**Reviewer Scores:**

NA

---

### Decision · Program_Chairs · 2026-01-26

Accept (Poster)